# Lung Inflammation Induced by Inactivated SARS-CoV-2 in C57BL/6 Female Mice Is Controlled by Intranasal Instillation of Vitamin D

**DOI:** 10.3390/cells12071092

**Published:** 2023-04-06

**Authors:** William Danilo Fernandes de Souza, Sofia Fernanda Gonçalves Zorzella-Pezavento, Marina Caçador Ayupe, Caio Loureiro Salgado, Bernardo de Castro Oliveira, Francielly Moreira, Guilherme William da Silva, Stefanie Primon Muraro, Gabriela Fabiano de Souza, José Luiz Proença-Módena, Joao Pessoa Araujo Junior, Denise Morais da Fonseca, Alexandrina Sartori

**Affiliations:** 1Department of Chemical and Biological Sciences, Institute of Biosciences, São Paulo State University (UNESP), Botucatu 18618-689, SP, Brazil; 2Laboratory of Mucosal Immunology, Department of Immunology, Institute of Biomedical Sciences, University of São Paulo (USP), São Paulo 05508-000, SP, Brazil; 3Laboratory of Emerging Viruses, Department of Genetics, Evolution, Microbiology and Immunology, Institute of Biology, University of Campinas (UNICAMP), Campinas 13083-862, SP, Brazil

**Keywords:** SARS-CoV-2, COVID-19, lung, inflammation, mice, vitamin D

## Abstract

The COVID-19 pandemic was triggered by the coronavirus SARS-CoV-2, whose peak occurred in the years 2020 and 2021. The main target of this virus is the lung, and the infection is associated with an accentuated inflammatory process involving mainly the innate arm of the immune system. Here, we described the induction of a pulmonary inflammatory process triggered by the intranasal (IN) instillation of UV-inactivated SARS-CoV-2 in C57BL/6 female mice, and then the evaluation of the ability of vitamin D (VitD) to control this process. The assays used to estimate the severity of lung involvement included the total and differential number of cells in the bronchoalveolar lavage fluid (BALF), histopathological analysis, quantification of T cell subsets, and inflammatory mediators by RT-PCR, cytokine quantification in lung homogenates, and flow cytometric analysis of cells recovered from lung parenchyma. The IN instillation of inactivated SARS-CoV-2 triggered a pulmonary inflammatory process, consisting of various cell types and mediators, resembling the typical inflammation found in transgenic mice infected with SARS-CoV-2. This inflammatory process was significantly decreased by the IN delivery of VitD, but not by its IP administration, suggesting that this hormone could have a therapeutic potential in COVID-19 if locally applied. To our knowledge, the local delivery of VitD to downmodulate lung inflammation in COVID-19 is an original proposition.

## 1. Introduction

SARS-CoV-2, a newly identified β-coronavirus, is the causative agent of the pandemic respiratory pathology known as COVID-19, whose peak occurred in 2020 and 2021. Even though most affected individuals are asymptomatic or develop mild symptoms, a minor proportion evolves towards a severe pathology. A plethora of factors related to the host, the environment, and the virus itself can affect the disease outcome [1]. Although the lung is considered the primary target of SARS-CoV-2, the virus can spread to many other organs such as the kidneys, intestine, liver, pancreas, spleen, muscles, and the nervous system [2,3]. Pulmonary manifestations vary from asymptomatic or mild pneumonia to a severe disease accompanied by hypoxia, shock, respiratory failure, and multiorgan deterioration or death [4]. The complexity of SARS-CoV-2 infection includes its aggravation by other comorbidities as hypertension, diabetes, and cardiovascular diseases [5] and by the adverse outcomes that may manifest after an acute illness and that are known as long COVID. In addition, there are emerging data on an extensive spectrum of sequelae associated with long COVID, mainly characterized by cardiovascular, pulmonary, and neuropsychiatric manifestations [6].

It is well established that the innate immune system works as the first line of response against pathogens, including SARS-CoV-2. This initial response is intended to limit viral infection and to promote the development of adaptive immunity. Pathogens, danger and damage-derived signals are detected by pattern-recognition receptors (PRRs) present in the surface, cytosol, or nucleus of epithelial cells, macrophages, monocytes, dendritic cells (DCs), neutrophils, and innate lymphoid cells (ILCs), which recognize PAMPs (pathogen-associated molecular patterns) and DAMPs (danger-associated molecular patterns). Several PRRs are able to mediate signaling pathways in response to an interaction with SARS-CoV-2 or to the products resulting from the viral infection, including Toll-like receptors (TLRs), retinoic acid-inducible gene-I-like receptors (RLRs), and nucleotide-binding oligomerization domain (NOD)-like receptors (NLRs). A detailed description of this interaction was recently published [7]. A growing body of clinical data have suggested that COVID-19 severity is mostly determined by inflammation and the associated cytokine storm [8,9]. The use of appropriate animal models allows a better understanding of infection and pathogenesis triggered by SARS-CoV-2. Most of the experimental in vivo studies have been conducted using macaques, cats, ferrets, hamsters, and mice, with hamsters and genetically modified mice being widely employed. Recently, it has been demonstrated that hamsters inoculated with SARS-CoV-2 by the intranasal (IN) route developed a viral pneumonia and systemic illness, showing histological evidence of lung injury, increased pulmonary permeability, acute inflammation, and hypoxemia [10].

Many of the findings described in mice are consistent with severe COVID-19 in patients. For example, the IN inoculation of SARS-CoV-2 in transgenic mice expressing the ACE2 receptor driven by cytokeratin-18 resulted in high virus levels in the lungs. An accentuated deterioration in the pulmonary function, which coincided with a local infiltration of monocytes, neutrophils, and activated T cells, was identified a few days later. Such inflammatory infiltrate displayed an impressive up-regulation of innate immunity markers, characterized by signatures of type I and II IFN and leukocyte activation pathways [11]. Standard laboratory mice strains and non-infectious virus components have also been used to establish models of lung inflammation. For instance, the intratracheal inoculation of SARS-CoV-2 N protein in C57BL/6, C3H/HeJ, and C3H/HeN mice induces an acute lung injury associated with inflammation through NF-kB activation [12]. Recently, a model of pulmonary inflammation induced by the lung coadministration of aerosolized SARS-CoV-2 spike (S) protein together with bacterial lipopolysaccharide (LPS) in C57BL/6 mice has also been described [13]. In particular, this procedure significantly increased the NF-kB activation, the number of inflammatory macrophages and polymorphonuclear cells (PMNs) in the BALF, and also triggered pathognomonic changes in the lungs. BALF analysis revealed an increased level of inflammatory cytokines and chemokines resembling a cytokine storm. In this context, the first objective of our investigation was to characterize the inflammatory lung process induced by the IN instillation of UV-inactivated SARS-CoV-2.

Most therapeutic strategies in clinical trials against COVID-19 consist of repurposing existing drugs already used for other infectious or inflammatory pathologies. Anti-viral drugs, monoclonal antibodies, high-titer convalescent plasma, and immunomodulators are frequently investigated [14,15,16]. Observational studies have shown that serum vitamin D (VitD) levels were inversely correlated with COVID-19 incidence and severity, suggesting that supplementation with this hormone could be explored to prevent or treat COVID-19 patients [17]. Since then, VitD has been tested, alone or associated with other pharmaceuticals, as a potential prophylactic, immunoregulatory, and even neuroprotective measure for this infection [18,19]. According to ClinicalTrials.gov, there are 31 completed studies involving tests with VitD in COVID-19 patients. Some of these trials aimed to assess the effects of VitD on the lungs indicated that one single dose did not prevent the respiratory worsening of hospitalized patients [20], nor did it reduce hospital length in moderate to severe COVID-19 [21]. On the other hand, other reports have been more promising, mainly by using multiple doses of this vitamin. For instance, multiple doses of VitD treatment have resulted in shorter lengths of stay, lower oxygen requirements, and a reduction in inflammatory markers status in COVID-19 patients [22]. Additionally, a 5000 IU daily supplementation for 15 days in VitD-deficient patients reduced the time to recovery for cough and gustatory sensory loss [23].

To the best of our knowledge, most of these trials were conducted by administering VitD orally, which is, considering some limitations, a route that allows a systemic drug distribution [24]. In this context, our second objective was to investigate if the lung inflammatory process induced by inactivated SARS-CoV-2 could be downmodulated by VitD administered by both intraperitoneal (IP) and IN routes. The choice of the IP route was based on our previous experience, showing that vitD was able to control the central nervous system (CNS) inflammation in an experimental murine model of multiple sclerosis [25]. The decision to test VitD administered via the IN route was adopted considering different reasons. Initially, we thought about practical issues as, for example, non-invasiveness, where there may be a possible immediate effect considering that VitD would be applied directly at the inflammatory site, and even the possibility of self-administration. We also considered the fact that previous reports indicate that VitD has a remarkable anti-inflammatory effect when locally applied to the respiratory system. This has already been demonstrated in some lung experimental conditions such as rhinitis [26] and asthma [27]. In addition, the in situ application of VitD has also been effective in other localized pro-inflammatory diseases, for example, vitiligo [28] and psoriasis [29]. The fact that IN VitD could theoretically control, at least partially, some of the immediate or late neurological alterations caused by the dissemination of SARS-CoV-2 to the nervous system was also pondered. This possibility was based on reports showing that VitD attenuates blood–brain barrier disruption [30], therefore decreasing the entry of inflammatory cells into the central nervous system. In addition, the nose-to-brain route has been proposed as a promising strategy for drug delivery to the brain [31].

## 2. Materials and Methods

### 2.1. General Experimental Design

In this investigation, we initially characterized a model of pulmonary inflammation induced by the intranasal (IN) administration of 3 doses of inactivated SARS-CoV-2 in C57BL/6 mice. Then, we evaluated the ability of VitD, administered by intraperitoneal (IP) (4 doses) or IN (3 doses) routes, to control or modify this process. The following methodologies were used: the total and differential count of cells in the broncho-alveolar lavage fluid (BALF), histopathological analysis of the target tissue, determination of lymphocyte subpopulations and inflammatory mediators by RT-PCR, flow cytometric analysis of cells recovered from the lung, and cytokine quantification in pulmonary homogenates. These analyses were performed on the seventh day after the administration of the first virus dose. The induction of lung inflammation and the evaluation of VitDs therapeutic potential are outlined in Appendix A—General experimental design, A and B, respectively, provided in the Appendix A. Body weight loss and serum calcium levels were also determined to assess the possible side effects of VitD.

### 2.2. Animals

Female C57BL/6 mice were acquired from the Animal Facility of the Animal Research and Production Center (ARPC/IBTECH), UNESP, Botucatu, or from the Animal Facility of the University of Sao Paulo. The animals were housed in polypropylene cages with a maximum capacity for 4–5 animals on a rack with individual ventilation (Alesco). The temperature was controlled by air conditioning and was maintained at about 22 °C. The animals received water and commercial feed ad libitum and were handled according to the standards of the ethics committee in animal experimentation of IB, UNESP, Botucatu (CEUA Protocol No. 1959140820, ID: 000129) and the ethics committee in animal experimentation of the ICB, USP (CEUA Protocol 3147240820).

### 2.3. SARS-CoV-2 Propagation and Inactivation

This study used a B lineage isolate of SARS-CoV-2 (SARS-CoV-2/SP02.2020, GenBank accession number MT126808) kindly provided by Edison Luiz Durigon (PhD, Institute of Biomedical Sciences–University of São Paulo–São Paulo-Brazil), recovered from a sample collected on 28 February 2020 in Brazil. The virus was propagated in Vero cells (CCL-81; ATCC, Manassas, VA, USA) according to the previously described protocol [32] in a biosafety level 3 laboratory (BSL-3) located in the University of Campinas. All the viral stocks used in the study were titrated using a plaque-forming assay according to previously published studies [33]. Briefly, decimal serially diluted samples were incubated with Vero cells into 24-well plates for 1 h at 37 °C and 5% CO_2_. After adsorption, the cells were overlaid and maintained with a semi-solid medium (1% *w*/*v* carboxymethylcellulose) in DMEM supplemented with 5% fetal bovine serum (FBS) for 4 days. After fixation with 8% formaldehyde solution and staining with 1% methylene blue (Sigma-Aldrich, St. Louis, MO, USA), the viral titer was determined by dividing the average number of plates by the value obtained from the multiplication between the dilution factor and the volume of the viral suspension added to the plate. The results were expressed as the viral plaque-forming units (PFU)/mL of the sample. The virus used in this study, with a titer of 8 × 10^6^ PFU/mL, was inactivated by exposure to 7560 mJ/cm^2^ of UVC (30 min) according to what has been described previously [34]. The supernatant of non-infected Vero cells, inactivated by UVC, was used as a negative control. The inactivation efficacy was determined by inoculating the UVC-inactivated product into Vero cells. The Vero cells infected with UVC-inactivated SARS-CoV-2 showed no cytopathic effect. In addition, no virus was detected in the supernatant of these cells by a plaque-forming assay or quantitative RT-PCR.

### 2.4. Induction and Characterization of Pulmonary Inflammation by SARS-CoV-2

We adopted a protocol which has been previously described [35]. Briefly, the animals received 3 doses of 4 × 10^5^ PFU/50 µL, administered on days 1, 3, and 5 and dispensed with a tip connected to a pipette. The pulmonary inflammatory response was analyzed on the 7th day by using 5 methodologies: total and differential cell counts performed in the BALF, RT-PCR for the quantification of the transcription factors, cytokines and inflammasome genes, flow cytometry for the identification of cells present in the parenchyma, histopathological analysis, and cytokine quantification in lung homogenates.

### 2.5. Bronchoalveolar Lavage Procedure

The bronchoalveolar washes were obtained from mice previously euthanized with ketamine and xylazine. The animal’s trachea was exposed with the help of scissors and tweezers, and a catheter was introduced through which 1 mL of sterile PBS was injected and then aspirated. This PBS injection/aspiration process was repeated 3 consecutive times and the samples were centrifuged at 4 °C for 10 min, 1500 rpm. The pellets were pooled and resuspended in 300 µL, and the total cell concentration was determined using a Neubauer chamber. Smears for differential cell counts were prepared by cytocentrifugation at 600 rpm for 5 min and then stained with the Rapid Pannotic Kit (Laborclin, Paraná, Brazil).

### 2.6. Quantitative Real-Time PCR (RT-qPCR) Analysis

The total RNA from the lung samples was extracted with the reagent TRIZOL (Invitrogen, Carlsbad, CA, USA) and the synthesis of cDNA (High-Capacity RNA-to-cDNA Converter Kit Applied Biosystems, Foster City, CA, USA), according to the manufacturer’s recommendations. The quantitative expression of mRNA for the transcription factors *Tbx21* (Mm00450960_m1), *GATA3* (Mm00484683_m1), *RORc* (Mm01261022_m1) and *Foxp3* (Mm00475162_m1), cytokines *IL*-*6* (Mm00446190_m1), *TNF*-*α* (Mm0043258_m1), *IFN*-*y* (Mm01168134_m1), *IL*-*12* (Mm00434169_m1), *IL*-*17* (Mm00439618_m1), inflammasome components as *NLRP3* (Mm09840904_m1), *IL*-*1β* (Mm00434228_m1), and *Caspase*-*1* (Mm00438023_m1), and other inflammatory markers *iNOS* (Mm00440502_m1), *CPA3* (Mm00483940_m1), and *Arginase* (Mm00475988_m1) were analyzed by real-time PCR, using the TaqMan system with primers and probes sold by Life Technologies (Applied Biosystems) according to the manufacturer’s recommendations. The gene expression was based on GAPDH (Mm99999915_g1), a reference gene, and presented as a relative change in the fold (2^−∆∆ct^), using the control group as a calibrator.

### 2.7. Lung Histopathological Analysis

Left lung samples were collected on the seventh day after the beginning of IN instillations and then they were washed with PBS, fixed in 10% buffered formalin for 24 h, and washed and stored in 70% ethanol until inclusion. Then, 5 µm thick sections from the control (saline), culture medium, and SARS-CoV-2 groups were obtained using a Leica RM2245 microtome and they were stained with H&E. Histopathological alterations were evaluated in a Carl Zeiss microscope GmbH, Oberkochen, Germany, attached to a digital camera (AxioCamHRc, Carl Zeiss, Oberkochen, Germany).

### 2.8. Isolation of Lung Cells and Flow Cytometry Analysis

In order to differentiate the parenchyma-infiltrating leukocytes from the vascular-associated fraction, the mice were intravenously injected with 3 µg of FITC-labeled anti-CD45 antibody (Biolegend, San Diego, CA, USA) in 200 µL of sterile saline solution. After 3 min, the mice were euthanized, and the lungs were perfused and collected for tissue processing. The vascular fraction of leukocytes was identified based on the anti-CD45 FITC staining and they were excluded from the analysis.

The right lungs, which were removed soon after euthanasia, were shredded, processed in digestion buffer (incomplete RPMI medium (Sigma, St. Louis, MO, USA)) containing 0.5 mg/mL of DNAse I (Sigma-Aldrich, USA) and 1 mg/mL of collagenase IV (Sigma Aldrich, USA) and incubated at 37 °C for 30 min at 180 rpm. Once homogenized, the digested samples were passed through 70 µm cell strainers, transferred to conical centrifuge tubes containing 8 mL of complete RPMI (3% FBS (Sigma-Aldrich, St. Louis, MO, USA)), 10 mg/mL of penicillin + 10,000 units/mL streptomycin (Hyclone, Logan, UT, USA), 0.3 g/mL of L-glutamine (Sigma-Aldrich, USA), 0.0040 g/mL of beta-mercaptoethanol (Sigma-Aldrich, USA), 0.0089 g/mL of non-essential amino acids (Sigma-Aldrich, USA), 0.0089 g/mL of sodium pyruvate (Sigma-Aldrich, USA), and then centrifuged at 4 °C for 8 min at 1600 rpm. The supernatants were discarded, and the cells were resuspended in 500 μL of ACK erythrocyte lysis buffer and incubated on ice. After 2 min, 10 mL of complete RPMI were added and the samples were centrifuged again at 4 °C for 8 min at 1600 rpm. Then the supernatant was discarded, and the cells were resuspended in 1 mL of complete RPMI, counted, and prepared for cytometry analysis. Two million lung cells were stained for surface markers or for transcription factors, according to Appendix A (available in the Appendix A). All the antibodies and intranuclear staining were conducted according to the manufacturer’s instructions using an eBioscience Transcription Factor Buffer set.

Alternatively, 2 million cells were used for the intracellular cytokine detection. For the labeling of cytokine-producing cells, the cells were incubated for 4 h with 100 μL of complete RPMI containing 50 ng/mL of phorbol myristate acetate (PMA) (Sigma Aldrich, USA), 500 ng/mL of ionomycin (Sigma-Aldrich, USA), and 1 μL/mL of GolgiPlug (BD Biosciences, San Jose, CA, USA). The cytokines, transcription factors, and cells from innate and specific immunity were then labeled with fluorochrome-conjugated antibodies. Prior to the addition of the antibody mix, as specified in Appendix A, all samples from all panels were incubated for 20 min at 4 °C with 30 μL of live dead, 1:1000 (LD, Thermo Fisher Scientific, USA), followed by surface staining and intracellular staining (BD-Citofix-Citoperm kit, USA). The data were acquired in the BD LSRFortessa X-20 flow cytometer (BD Biosciences, USA) and the compensation and data analyses were performed using the FlowJo software. The gate strategies are described in Appendix A.

### 2.9. VitD Administration by IP and IN Routes

1α,25-dihydroxyvitamin D3 (1,25-VitD3, Sigma-Aldrich, USA) was administered by IP or IN routes. The 2 therapeutic protocols with 1α,25-dihydroxyvitamin D3 (VitD) were carried out using different strategies. In the IN protocol, each animal was treated with 3 doses of VitD (0.1 µg/dose), which were administered simultaneously with the SARS-CoV-2 inoculum (4 × 10^5^ PFU/each inoculum) in a final volume of 57 μL. This volume was divided between the 2 nostrils on days 1, 3, and 5. In the IP protocol, each animal was treated with 4 doses of VitD (0.1 µg/100 µL/dose) that were delivered on days 0, 2, 4, and 6 to mice that were instilled with 50 µL of SARS-CoV-2 on days 1, 3, and 5. In both cases, euthanasia was performed at the seventh day after the beginning of the protocol.

### 2.10. Measurement of Serum Calcium Levels

The blood samples collected after anesthesia were centrifuged, and the sera were stored at −20 °C until further analyses. The serum levels of calcium were measured according to the instructions of the manufacturer (Cálcio Arsenazo III, Bioclin-Quibasa Química Básica Ltda, Belo Horizonte, MG, Brazil). In this technique, calcium quantification was based on a colorimetric reaction in which calcium reacts with arsenazo III, in an acidic medium, generating a blue complex whose intensity is proportional to the calcium concentration in the sample.

### 2.11. Statistical Analysis

In the case of parametric variables, the values were presented as the mean and standard error of the mean (SEM), and the comparison between the two groups was performed using an unpaired *t*-test and, among three or more groups, an ANOVA was performed followed by Tukey’s test. When the variables were non-parametric, the results were presented in median and interquartile intervals and the comparison between the groups was performed using Kruskal–Wallis’ test followed by Dunn’s test. The level of significance adopted was 5%. The data were analyzed using the SigmaPlot for Windows version 2.0 statistical package (1995, Jandel, Corporation, CA, USA). For t-distributed stochastic neighbor embedding (t-SNE) algorithm analysis, 100,000 or 50,000 events per sample, were downsampled from the live parenchymal leukocytes gate (Appendix A) and concatenated. The t-SNE algorithm was applied in the concatenated samples using 2000 interactions and perplexity 80. After that, the cell clusters were identified based on the main cell subsets gated according to Appendix A, and the percentage of each cell subset was calculated after segregating the groups based on the sample IDs.

## 3. Results

### 3.1. Cell Infiltration in the BALF Suggests Pulmonary Inflammation in Mice Intranasally Instilled with Inactivated SARS-CoV-2

The ability of inactivated SARS-CoV-2 to trigger a lung inflammatory process was initially investigated by analyzing the amount and identity of white blood cells (WBCs) obtained from the broncho-alveolar lavage fluid (BALF). Two control groups were included in all the initial experimental procedures and were identified as the control and culture medium, which corresponded to animals that were anesthetized and instilled with 0.9% saline or with the culture medium used for virus propagation in VERO cells, respectively. The total number of WBCs and specific cell populations were identified in cytospin smears, are shown in Figure 1A, and they indicate a significant increase in the total cell number, as well as in lymphocytes and neutrophils in animals that received SARS-CoV-2 in comparison to the control groups. The percentage alterations observed in the SARS-CoV-2 group included a significant decrease in macrophages and a significant increase in neutrophils, as illustrated in Figure 1B. This lower percentage of macrophages in the SARS-CoV-2 group, in comparison to the control groups, indicates an increment of other cell types as lymphocytes (discreet) and PMNs (significant) associated with the cellular influx to the lungs triggered by the virus. Even though the proportion of macrophages was smaller, the total number of this cell type was almost double in the SARS-CoV-2 group, in comparison to the control groups (Figure 1A). Animals injected with saline or culture medium displayed a similar profile, characterized by a smaller number of all cell types, indicating that the culture medium present in SARS-CoV-2 preparation was not triggering a significant pulmonary airway inflammation.

### 3.2. RT-qPCR from Lung Homogenates Shows Alterations in T Cell Subsets, Cytokines, and Other Inflammatory Mediators

Next, we measured the relative expression of several genes by RT-q PCR which revealed differences between culture medium and SARS-CoV-2 groups. In the SARS-CoV-2 group, there was a significantly higher expression of *Foxp3* (Figure 1F), *IL*-*6* (Figure 1G), and *GM*-*CSF* (Figure 1L) transcripts and a higher, even though not statistically significant, expression of *TNF*-*α* (Figure 1H), *IL*-*17* (Figure 1K), *IL*-*1β* (Figure 1O), and *NLRP3* (Figure 1P) transcripts. On the other hand, we found a significant decrease in the expression of *RORc* (Figure 1E) and *iNOS* (Figure 1M) in the lungs of the SARS-CoV-2 group compared to the culture medium control. Other genes, such as *T*-*bet*, *GATA*-*3*, *IFN*-*γ*, IL-12, and *CPA3* (Figure 1C,D,I,J,N, respectively), were similarly expressed in the culture medium and SARS-CoV-2 groups.

### 3.3. Histopathology and Cytometric Analysis Reveal an Impressive Infiltration of Inflammatory Cells into the Pulmonary Parenchyma

According to the histopathological evaluation illustrated in Figure 2A, the lung architecture was preserved in the animals from the control group, allowing the visualization of alveoli and longitudinally and transversally sectioned blood vessels and bronchi. The culture medium and SARS-CoV-2 groups displayed inflammatory foci; however, the ones found in the virus-instilled animals were clearly more numerous and intense. These inflammatory foci were, in both cases, located around the vessels and bronchi, as indicated by black and green arrows, respectively. The presence of neutrophilic infiltrates (green arrowhead), macrophage infiltrates (blue arrowhead), and lymphocytic infiltrates (yellow arrowhead) are indicated in the microphotographs. Numerous consolidation areas were present in the lungs of SARS-CoV-2-instilled animals, but they were rare and absent in the culture medium and saline control groups, respectively. To confirm these findings, we evaluated the leukocytes infiltrating the lung parenchyma by using cells isolated from mice previously injected with FITC-labeled anti-CD45 antibodies to distinguish circulating cells from the tissue infiltrate. Indeed, confirming the histopathological findings, the quantification of total and parenchymal infiltrating CD45^+^ leukocytes revealed a significant increase in the cell number in the group of mice that received UV-inactivated SARS-CoV-2 compared to both control groups (Figure 2B).

Next, we characterized the tissue-infiltrating leukocytes based on the surface molecules expression using flow cytometry, and the t-SNE algorithm was used for dimension reduction (Figure 2C and Appendix A). In the lungs of mice exposed to the inactivated virus, we found an enrichment in the clusters of cells that indicate the presence of neutrophils, eosinophils, macrophages (tissue-resident), and monocytes in the CD103^-^CD11b^+^ DC subset (Figure 2C). On the other hand, the frequency of other cell subsets, such as CD103^+^CD11b^-^ DCs, alveolar macrophages, and B cells, were reduced (Figure 2C). Furthermore, the frequency of proinflammatory cytokine-producing cells was also increased in the SARS-CoV-2 group, in particular, the frequency of IFN-*γ*- and TNF-α-producing TCRβ^+^ T cells and IL-6- and TNF-α-producing CD11b^+^ myeloid cells (Figure 2D). The quantification of each cell subset number is shown in Figure 2F–Q. The predominant profile of all the tested cells was characterized by a significantly higher number of cells enriched in the t-SNE analysis in the SARS-CoV-2 group in comparison to the control group. This was the case for the total number of neutrophils (Figure 2F), eosinophils (Figure 2G), inflammatory monocytes (Figure 2I), resident macrophages (Figure 2K), dendritic cells (DCs) (Figure 2L), CD11b^+^CD103^-^ DCs (Figure 2M), IFN-*γ*^+^- and TNF-α^+^-producing T cells (Figure 2N–O and Appendix A), and TNF-α^+^- and IL-6-producing myeloid cells (Figure 2P–Q and Appendix A). Contrastingly, the total number of each cell population in the culture medium-instilled animals was intermediate between the control and SARS-CoV-2 groups (data not shown). Therefore, these data show that the IN instillation of the inactivated virus is sufficient to promote a proinflammatory lung milieu that resembles most of the markers of the infection with SARS-CoV-2.

### 3.4. BALF and Cytokine Levels in Lung Homogenates Suggest That in VitD Modulates Pulmonary Inflammation Induced by SARS-CoV-2

Considering that VitD has a strong effect on the immune system and that it is considered for prophylactic or therapeutic application in COVID-19 patients [18,19], we tested its IP and IN effectiveness to control experimental lung inflammation induced by inactivated SARS-CoV-2. As already observed in previous studies, IP VitD administration triggered a significant loss of body weight (Figure 3A) and also significant hypercalcemia (Figure 3B) in comparison to all the other experimental groups. Even though VitD also significantly increased serum calcium levels, it only slightly increased body weight loss, as illustrated in Figure 3A,B, respectively. Concerning the BALF, the comparison among SARS-CoV-2, SARS-CoV-2/VitD (IP), and SARS-CoV-2/VitD (IN) showed no differences in the total number of WBCs, macrophages, and lymphocytes. However, a significant decrease in PMNs was detected in the SARS-CoV-2/VitD (IN)-treated group in comparison to the non-treated groups. Additionally, the number of eosinophils in the SARS-CoV-2/VitD (IP) group was significantly higher than in the IN-treated one. Concerning the percentage of these cells, a higher percent of macrophages was found in the IN-treated group in comparison with the two other groups, and a decreased percentage of PMNs and eosinophils was observed in the IN-treated group in relation to the non-treated one. The percent of eosinophils in the IN-treated mice was also significantly reduced in comparison to the IP-treated ones. To analyze if the SARS-CoV-2-induced lung inflammation model was also mimicking the cytokine storm-like phenomenon and to reinforce the presumed down-modulatory effect of VitD, we tested the presence of pro-inflammatory and regulatory cytokines in lung homogenates. As can be observed in Figure 3E, IN VitD decreased TNF-α and IL-6 and IP VitD decreased IL-6 levels; however, these alterations were not significant. No changes were detected in IL-17A and IFN-γ (Figure 3E) or in the other tested cytokines as IL-2, IL-4, and IL-10 (data not shown).

### 3.5. Differential Effects of VitD Delivered by IN and IP Routes on RORc and Inflammasome Genes Expression

The expression of various genes in the lungs of mice IN challenged with UV-inactivated SARS-CoV-2 was similar in the three compared groups, as was the case of *T*-*bet*, *GATA3*, *Foxp3*, *TNF*-*α, IFN*-*γ*, *IL*-*12*, *GM*-*CSF*, and *INOs* (Figure 4A,B,D–J, respectively). In contrast, the expression of *RORC* (Figure 4C) was significantly higher in the IP VitD-treated mice compared to the SARS-CoV-2 group, and the expression of *IL*-*1β*, and *NLRP3* was significantly higher in the IP VitD-treated group in comparison to the IN VitD-treated one (Figure 4C,K,L), respectively. A significantly reduction in the *ARG* expression was observed in the SARS-CoV-2/VitD (IN) group in comparison to the SARS-CoV-2 group (Figure 4J). In order to clarify whether these slight changes in gene expression would be reflected in the inflammatory infiltrate, we next performed flow cytometry of tissue infiltrating cells to better define the immunological changes associated with VitD treatment.

### 3.6. IN VitD Treatment Efficiently Controls Pulmonary Inflammation

The analysis of the histological sections clearly indicates the strong ability of IN VitD, in contrast to IP VitD, to control lung inflammation. In this case, there was a convincing interruption of the accumulation of inflammatory cells in the lung parenchyma of IN VitD-treated mice that were exposed to the inactivated virus. These findings can be clearly observed in Figure 5A. Inflammatory foci are easily observed around the vessels (black arrows) and bronchi (green arrows) in SARS-CoV-2 and SARS-CoV-2/VitD IP groups. The presence of neutrophilic (green arrowhead), macrophage (blue arrowhead), and lymphocytic infiltrates (yellow arrowhead) are also numerous in these two experimental groups, but rare in animals treated with VitD through the IN route.

To clarify whether this anti-inflammatory effect of IN VitD was involved in the modulation of specific cell types, including myeloid, DCs, ILCs, and lymphocytes, we used flow cytometry to identify the cells infiltrating in the lung parenchyma, as described above. The total number and the frequency of CD45^+^ cells infiltrating the lung parenchyma of mice instilled with inactivated SARS-CoV-2 was significantly increased compared to both control groups (control and culture medium) (Figure 5B,C). Notably, the IN VitD treatment significantly reduced the number and frequency of leukocytes infiltrating the lung parenchyma of mice receiving the inactivated virus (Figure 5B,C).

Next, to better understand the modulatory effects of VitD in the virus-induced parenchymal lung inflammation, we analyzed the frequency of distinct cell subsets by t-SNE and found that the VitD treatment reverted the recruitment of neutrophils, eosinophils, patrolling monocytes, and CD103^-^CD11b^+^ DCs induced by the inoculation of inactivated SARS-CoV-2 (Figure 5D,E and Figure 6C). Notably, the VitD treatment increased the percentage of B cells and CD103^+^CD11b^-^ DCs in the lung parenchyma in comparison to the SARS-CoV-2 group (Figure 5E and Figure 6C), suggesting that the treatment might be selectively controlling the inflammatory immune tone in the lung.

In addition, the functional impact of an IN VitD treatment on inflammatory cytokine production was analyzed using flow cytometry in myeloid, B, Tγδ, ILCs, and TCD4^+^ cells (Figure 6). Concerning cytokine production by myeloid cells, while the inactivated virus promoted the production of TNF-α and IL-6 by myeloid lung cells, the IN VitD treatment controlled the frequency of cytokine-producing cells (Figure 6A). When we quantified the number of cytokine-producing myeloid cells, the only population displaying a significant difference was the one producing both cytokines. Even though the percentage of TNF-α^+^IL-6^+^CD11b^+^ cells was similar in SARS-CoV-2 and SARS-CoV-2/VitD groups (Figure 6E), the total number of these cells was significantly lower in the VitD-treated animals, as shown in Figure 6D. These data could be explained by the consistent reduction in total cell recruitment to the lung parenchyma in the VitD-treated mice.

The frequency of DCs was only slightly reduced by VitD (data not shown). As already observed during the characterization phase of the lung inflammatory process, the amount of DCs was very similar in SARS-CoV-2 and its respective control group (culture medium). In spite of this, VitD therapy was able to significantly reduce the total number of DCs (Figure 6F) and also of the two evaluated subsets, CD103^-^CD11b^+^ (Figure 6G) and CD103^+^CD11b^-^ (Figure 6H), but not the frequency of CD103^+^CD11b^-^ (Figure 5B,D). Once again, this inconsistency in the modulation of the cell number but not in its frequency could be attributed to the considerable reduction in leukocyte recruitment to the lung parenchyma of VitD-treated animals (Figure 5B).

The total numbers of IL-17- and IL-6-producing ILCs were usually increased in the SARS-CoV-2 group in comparison to the control that received the culture medium, and VitD therapy triggered a clear tendency to decrease the total number of theses cytokine-producing cells, as shown in Figure 6A,J,K. Concerning the Tγδ lymphocytes, the most relevant alterations were detected in the cells that were producing IL-17 or IFN-γ. As shown in Figure 6A,L,N, VitD significantly downregulated the total cell number of IL-17- and IFN-γ-producing cells. VitD also downmodulated, although not significantly, their percentages, as illustrated in Figure 6M,O, for IL-17 and IFN-γ, respectively. In regard to TCD4^+^ lymphocytes, the most pronounced differences were also observed in IL-17- and IL-17/IFN-γ-producing cells. The % of these cells was reduced by treatment with VitD, making this reduction statistically significant in the case of TCD4^+^IL17^+^ (Figure 6A,Q). The total amount of these two cell subsets was also decreased by VitD therapy, making this reduction statistically significant regarding TCD4^+^IL-17^+^IFN-γ^+^ (Figure 6R). The % and the total number of B cells were similar in the culture medium and SARS-CoV-2 groups (Figure 6T,U). However, a significant increase in these two parameters was triggered by a local IN VitD administration, as shown in figures T and U. Interestingly, the number of IL-6^+^TNF-α^+^ B cells, which was increased in the SARS-CoV-2 group, was significantly downregulated by IN VitD (Figure 6V).

Taken together, our data suggest that the local administration of VitD was sufficient to suppress the recruitment of inflammatory cells to the lung parenchyma induced by the exposure to inactivated SARS-CoV-2.

## 4. Discussion

This investigation was conducted considering that COVID-19 can be a lethal disease and which the treatment for is not well established. Initially, we used female C57BL/6 mice, instilled with UV-inactivated SARS-CoV-2, to establish a working model of inflammation in the lung, the initial and main target of COVID-19 [36]. We then employed this model to investigate the potential of VitD to control local inflammation. The choice of the C57BL/6 mice strain and inactivated virus would, in our view, make the model more accessible to a greater number of researchers and laboratory facilities and allow for a further use of transgenic mice to answer specific questions about the inflammatory response during SARS-CoV-2 infection.

The initial results obtained by analyzing the cell influx to the BALF revealed the ability of inactivated SARS-CoV-2 to trigger a local inflammatory process characterized by an increase in WBCs, including in lymphocytes and neutrophils. As the BALF obtained from the culture medium group presented a profile very similar to the other control group (saline), most of the inflammatory process can be attributed to the virus and not to the content of the medium used to grow the virus. Altogether, the total number of WBCs along with their different cell types presented a clear and more direct idea of the inflammatory extension. On the other hand, the calculation of the percentage of each cell type presented an idea of the pattern of the immune response recruited to this compartment. In our experimental model, the immune tone of the airways was shifted towards a neutrophilic inflammatory profile in the detriment of a mononuclear or eosinophilic infiltrate. Therefore, we found an increase in both the percentage and total counts of neutrophils, but not in the other cell subsets.

Even though the subsequent analyses provided much more enlightening information about this model, these preliminary data were considered relevant because BALF procedures have been largely employed as a tool to study a plethora of experimental and human lung diseases [37,38]. In addition, this technique has been explored in experimental and clinical investigations involving the SARS-CoV-2 virus itself [39].

The analysis of the lung RNA expression reinforced the initial findings, showing an increased expression of *GM-CSF* and *Foxp3* and a tendency towards increased values for *IL-17*, *IL-1β,* and *NLRP3* mRNA expression. The possible contribution of the inflammasome activation to COVID-19 immunopathogenesis is highly supported by the literature. It has been reported that inflammasome activation is triggered by SARS-CoV-2 components [40,41], that its higher activation is possibly involved in COVID-19 severity [42], and that the specific inhibition of the NLRP3 inflammasome was able to decrease the intensity of a COVID-19-like pathology in mice [43]. In addition, NLRP3 inflammasome activation during COVID-19 can also be induced by DAMPs released as a result of the initial innate inflammatory process that follows the exposure to SARS-CoV-2 components [44]. For instance, the inflammatory process that drives cell damage and extracellular ATP release can activate the purinergic P2X7 receptor, resulting in K+ efflux and, consequently, NLRP3 inflammation [44,45]. Notably, the inflammasome activation throughout this process does not require the active infection of the virus, but this could be induced by the inflammation resulting from viral components exposure. Considering this scenario, it seems plausible to hypothesize that this could be one of the pathways for NLRP3 activation in our inflammation experimental model triggered by inactivated SARS-CoV-2 instillation. Another possibility for NLRP3 activation by viral components could be the accumulation of angiotensin II in the cell which results from the interaction of SARS-CoV-2 protein with ACE2 in the cell surface. This process reduces angiotensin II degradation and its subsequent accumulation in the cell [44].

Histopathological analysis, together with the flow cytometry analysis of the cells obtained from the lung parenchyma, allowed a better evaluation of the intensity and quality of the inflammatory process triggered by the exposition to the virus. H&E-stained sections clearly showed that the IN instillation of inactivated SARS-CoV-2 induced a multifocal and interstitial pneumonia characterized by perivascular and perialveolar inflammation. Flow cytometric evaluation performed with the cells isolated from the lung parenchyma allowed a more precise identification of the cells involved in local inflammation. A plethora of cell types, such as PMNs, eosinophils, lymphocytes, and macrophages, including monocyte-derived macrophages and parenchyma-resident macrophages, were identified using this methodology [46]. All these cellular types have been associated with COVID-19, and their contribution to disease immunopathogenesis has been apprised in pre-clinical and clinical studies [39,40,41]. An increased amount of PMNs is described in the bloodstream and the lungs of COVID-19 patients, and strong evidence indicates that they play a paramount role in disease pathophysiology [47]. A neutrophilic mucositis involving the entire lower respiratory tract has been described in lung autopsies from COVID-19 deceased patients [48]. Moreover, a neutrophil activation signature predicted critical illness and mortality in COVID-19 [49]. Most of the damage triggered by PMNs has been attributed to their extensive and prolonged activation, which leads to an excessive ROS release composed of superoxide radicals and H_2_O_2_ [50]. In addition, according to [51], PMNs have been seen as drivers of hyperinflammation by enhanced degranulation and pro-inflammatory cytokine production. The release of neutrophil extracellular traps (NETs) by PMNs is also pointed as a major promotor of damage in COVID-19 by causing endothelial injury and necroinflammation via complement activation, and by promoting the formation of venous thrombi [52]. This activation of PMNs could be directly determined by the virus. It was recently described [53] that single-strand RNAs from the SARS-CoV-2 genome are able to activate human neutrophils via TLR8, triggering a remarkable production of TNF-α, IL-1ra, and CXCL8, apoptosis delay, the modulation of CD11b and CD62L expression, and the release of NETs. Additionally, the tissue damage induced by the neutrophilic infiltration can activate the inflammasome, as described above, resulting in more inflammation and neutrophil recruitment, perpetuating, therefore, the inflammatory process. This exuberant contribution of PMNs to the interstitial pneumonia that occurs in COVID-19 was, in many aspects, reproduced in an h-ACE2 mouse model infected with SARS-CoV-2 [54].

The presence of dendritic cells (DCs) in the pulmonary parenchyma also deserves attention considering that they are fundamental for both an innate and specific anti-viral immune response, but they can also contribute to viral dissemination and immunopathogenesis during COVID-19 [55]. In this regard, by analyzing circulating DCs and monocyte subsets from hospitalized COVID-19 patients, [56] described their impaired function and delayed regeneration. Flow cytometry also allowed the identification of lymphoid and myeloid cells producing cytokines such as TNF-α, IFN-γ, and IL-17, which are among the most important mediators of COVID-19 immunopathogenesis [57].

The validation of our model as an adequate tool to investigate other procedures to control lung inflammation is supported by another investigation ongoing in our research group. The histological changes that we found after the instillation of the UV-inactivated SARS-CoV-2 are comparable to the lung inflammation that h-ACE2 mice develop after the active infection. The profile of inflammatory cells eluted from the lung parenchyma is also very similar to the one described in our investigation (Aype et al., unpublished data).

This validation is also reinforced by the data described by [58]. These authors developed a model of SARS-CoV-2-induced acute respiratory distress syndrome by the intratracheal instillation of formaldehyde-inactivated SARS-CoV-2. Their described histopathological alterations and profile of cells infiltrated in the lungs are also similar to our findings.

Having confirmed that SARS-CoV-2 IN instillation triggered a pulmonary inflammation similar to that developed by the instillation of active or inactivated SARS-CoV-2 in h-ACE2 transgenic mice, our model employing UV-inactivated SARS-CoV-2 was used to test the ability of VitD to modulate the lung inflammatory process. The option for VitD was based on the extensive literature, attesting the powerful immunomodulatory property of this hormone [59], the robust evidences linking its low levels with poor COVID-19 outcomes [60], and our own previous experience, indicating its ability to counteract the inflammatory process that damages the central nervous system (CNS) in a multiple sclerosis (MS) murine model [25,61]. As indicated by the results, only IN VitD was capable of controlling pulmonary inflammation by downmodulating the presence of proinflammatory cytokine-producing cells. The effectiveness of IN VitD was confirmed by histopathological and flow cytometry analyses. The H&E sections from these animals revealed well-preserved lung structures, similar to those observed in the animals from the control group which were instilled with saline. The flow cytometry analysis indicated that, in this case, VitD was able to impair the recruitment of several cell types, as neutrophils, DCs, and lymphocytes, such as TCD4^+^ and Tγδ, in the lungs of mice challenged with SARS-CoV-2. This approach also allowed the identification of various cell subsets whose cytokine production was decreased by VitD, including myeloid (CD11b+), ILCs, Tγδ, TCD4^+^, and B cells. These findings were considered especially relevant because the main detected cytokines, TNF-α, IL-6, IL-17, and IFN-γ, have been identified as some of the major villains of the cytokine storm associated with COVID-19 severity and were significantly downmodulated by VitD.

The possible association between COVID-19 and VitD levels has been investigated from different perspectives, including the possible role of its deficiency and worst disease outcomes [62] and its prophylactic, immune regulatory, and protective role in COVID-19 [19]. Its therapeutic benefit is also being widely pursued, but a final conclusion is not possible yet due to the discordant results reported so far [63,64]. As far as we know, there are no publications concerning the administration of IN VitD to control lung inflammation triggered by SARS-CoV-2 in animal models or patients up to now. In this context, and considering the efficacy of its IN instillation demonstrated here, we believe that once this effect had also been proven in SARS-CoV-2-infected animals, it would be worth going to clinical trials.

Our initial hypothesis predicting a superior efficacy of IN VitD was based, among other information, on the fact that other lung inflammatory pathologies, such as experimental asthma and rhinitis, were efficiently controlled by local (IN) vitD delivery [26,27]. We also considered the fact that it is increasingly recognized that local synthesis of active VitD is more relevant for many of its immune effects on respiratory diseases than its systemic production [65]. We did not investigate in detail the mechanism by which the IN route, in contrast to the IP one, effectively controlled lung inflammation. We could speculate that the IN protocol, which theoretically allows the local availability of VitD during the initial interaction of the virus with pulmonary immune cells, could decrease the intensity of this interaction by, for example, locally decreasing the TLR expression. This effect, which has already been demonstrated after the exposition of peripheral blood mononuclear cells to VitD, decreased the production of pro-inflammatory cytokines [66]. We could also theorize that local VitD instillation is in the lung-draining lymph nodes and in the lungs themselves considering that this is one of the goals of local drug delivery [67]. However, future studies are necessary to measure the local vitD bioavailability and the optimal dose–response kinetics following its IN administration.

Based on the literature, we expected results supporting the more classical mechanisms attributed to immunomodulation by VitD as an induction of tolerogenic DCs [65], the expansion of Tregs (Ma et al., 2021), and reduced Th1/Th17 polarization [68]. Even though these canonical mechanisms were not observed, the cytometry results clearly showed the reduced production of pro-inflammatory cytokines by different cell types, which has been considered a relevant mechanism by which VitD could control elicitation and resolution phases of acute inflammation [19]. A possible reduction in the TLR expression, as proposed above, could additionally decrease the initial tissue damage by the early blocking of the release of chemokines, and therefore control the subsequent movement of leukocytes towards the lung. In line with this hypothesis, VitD is also capable of inhibiting NLRP3 inflammasome [69]. As discussed before, NLRP3 activation could be one of the main drivers for the innate inflammation after the virus exposure. Indeed, we found a reduction, even though not statistically significant, in the IL-1β expression in the lungs of IN VitD-treated animals. In addition, this treatment reduced the production of the inflammatory cytokines that we have evaluated using flow cytometry and could initiate the leukocyte influx to the lung tissue. In this context, we cannot exclude a direct impact of VitD in the lung epithelial mucosa [70]. Possibly, by interfering in the initial response of the epithelial cells to the interaction with the virus, IN VitD could control the initial release of chemokines and cytokines that will initiate the inflammatory loop driven by the virus. Therefore, we strongly believe that VitD is blocking the initial innate signals that drive the influx of inflammatory cells to the lung parenchyma instead of reversing or suppressing an already established inflammation. Notably, as stated before and supporting this hypothesis, we found no increase in Tregs or IL-10 production in the lungs of IN VitD-treated mice. In addition to the blockage of inflammasome activation, classical immunomodulatory mechanisms involving the innate immunity as the inhibition of DC maturation and blockage of antigen presentation to T helper cells could also occur. In addition, VitD suppresses the release of a plethora of pro-inflammatory cytokines [19], which seem, considering our results, to play a major role in its therapeutic effect when delivered intranasally. The model of inflammation limited to the lung, used in this work, does not allow us to predict whether IN VitD would control extrapulmonary inflammatory processes triggered by SARS-CoV-2 infection. As IN VitD was able to attenuate LPS-induced acute lung inflammation [71], it is expected that its application by this route would also be effective to control the pulmonary inflammatory processes triggered by other infectious agents or substances.

Even though the IP administration of VitD triggered a few downmodulatory effects, this procedure was not able to control lung inflammation. Conversely, this protocol increased the IL-1β, NLRP3, and RORc expression, suggesting a possible toxic proinflammatory activity associated with an excess of VitD. Actually, some authors have raised the possibility that VitD excess could trigger inflammation through T-cell stimulation via hypercalcemia. In this sense, serum calcium levels and body weight loss have been frequently employed to indicate VitD toxicity [72,73]. In healthy individuals, exogenous VitD toxicity is generally associated with the continuous use of high VitD doses [74]. Even though only a few VitD doses were employed in our protocols, calcium levels were similarly altered in IP- and IN-treated mice, possibly excluding the extracellular hypercalcemia in IP VitD-treated mice as the cause of inflammasome activation [75]. Of note, the IP VitD-treated animals also lost significantly more weight than the ones treated by IN VitD. If this accentuated body weight loss, which is also indicative of VitD toxicity, is somehow related to IP VitD ineffectiveness in controlling lung inflammation, it is not known yet. As body weight loss during VitD treatment has been attributed to its effect in the brain [63], a simple explanation for the finding that VitD IP causes much more weight loss than IN VitD is that IP VitD determines a higher concentration of this vitamin in the brain. A pharmacodynamic study of the tissue distribution of VitD administered by these two routes, especially in the CNS and in the lungs, will be necessary to understand this differential effect.

We believe that the most relevant contribution of this investigation is the proof of concept that IN VitD can significantly control the lung inflammatory process triggered by the local presence of the virus. Our study seems to be the first report suggesting that IN VitD administration has the potential to control inflammation induced by viral components. Future studies are indeed required to compare the efficacy in relation to the oral route, to define a better dose–response, and also to understand the pharmacokinetics and possible reduction in systemic side effects associated with both delivery routes. In addition, we have already observed that inflammation triggered by viable SARS-CoV-2 closely resembles the one induced by the inactivated virus (manuscript in preparation). The efficacy of VitD to control inflammation during an active SARS-CoV-2 infection requires a future and careful investigation and will possibly demand the association to virucidal drugs.

Even though the focus of our work has been the control of lung inflammation, we conceive that the possible adoption of IN VitD could bring additional advantages to COVID-19 patients. In this sense, we highlight the stabilizing activity towards the blood–brain barrier (BBB) disruption and the anti-fibrotic property of VitD considering that an increased BBB [76] and lung fibrosis had been associated with more severe COVID-19 cases.

Our study is mainly limited by the fact that we did not show that this anti-inflammatory effect of VitD also occurs during experimental SARS-CoV-2 infection. However, considering its adjunct therapeutic potential for COVID-19, we understand that this anti-inflammatory activity determined by IN VitD deserves to be further and fully investigated in preclinical and clinical assays.

## 5. Conclusions

The results provided by our investigation suggest a promising potential of VitD delivery by the IN route to control the pulmonary inflammation associated with the presence of SARS-CoV-2 antigens/components in the lungs. Further preclinical and clinical investigations will be essential to determine if these experimental findings can be translated to SARS-CoV-2 infection in humans.

## Figures and Tables

**Figure 1 cells-12-01092-f001:**
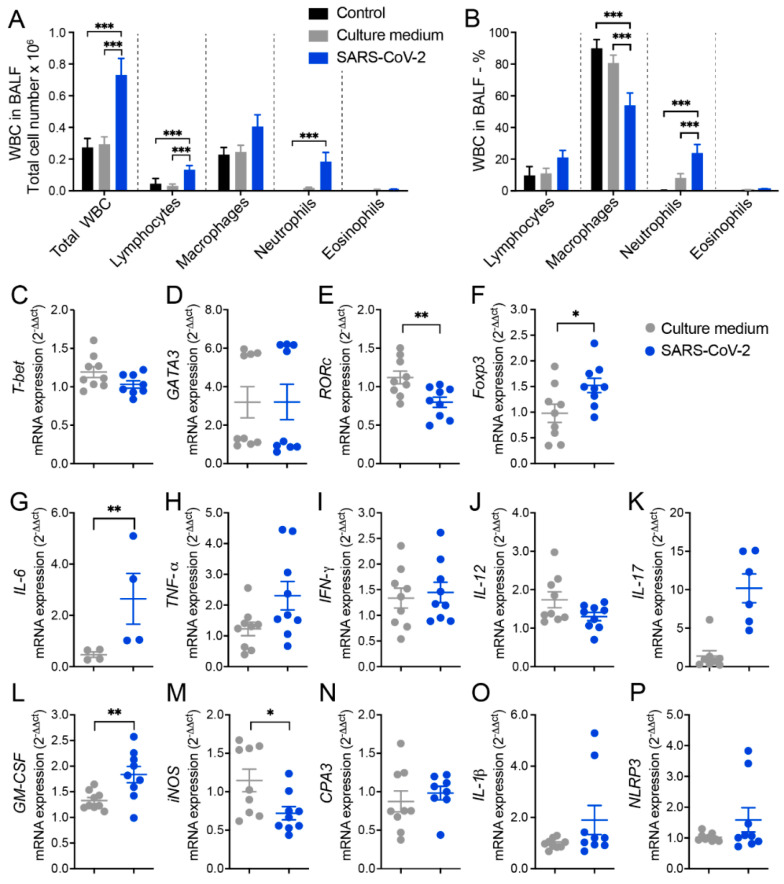
Cell counts in the BALF and lung mRNA transcripts for T cell subsets, cytokines, and other indicators of inflammation in mice intranasally instilled with UV-inactivated SARS-CoV-2. C57BL/6 mice were instilled with the virus (3 doses of 4 × 10^5^ PFU/each) on days 1, 3, and 5. On the 7th day, the BALF and the left lower lobe were collected for WBCs differential count and mRNA transcript determinations, respectively. Total number (**A**) and percentage (**B**) of WBCs: *T*-*bet* (**C**), *GATA3* (**D**), *RORc* (**E**), *Foxp3* (**F**), *IL*-*6* (**G**), *TNF*-*α* (**H**), *IFN*-*γ* (**I**), *IL*-*12* (**J**), *IL*-*17* (**K**), *GM*-*CSF* (**L**), *iNOS* (**M**), *CPA3* (**N**), *IL*-*1β* (**O**), *NLRP3* (**P**). In figure (**A**,**B**), the results are expressed as mean ± SEM and the statistical significance of the differences was analyzed using ANOVA followed by Tukey’s test. In figures (**C**–**P**), the results were expressed in median and interquartile intervals and the comparison between the groups was performed using the *t*-test. Data shown in (**A**,**B**) and (**C**–**P**) are derived from two experiments with similar results which were combined (n = 9), except *Tbet* (n = 8), *IL*-*17* (n = 6), and *IL*-*6* (n = 4). * *p* < 0.05; ** *p* < 0.01; *** *p* < 0.001.

**Figure 2 cells-12-01092-f002:**
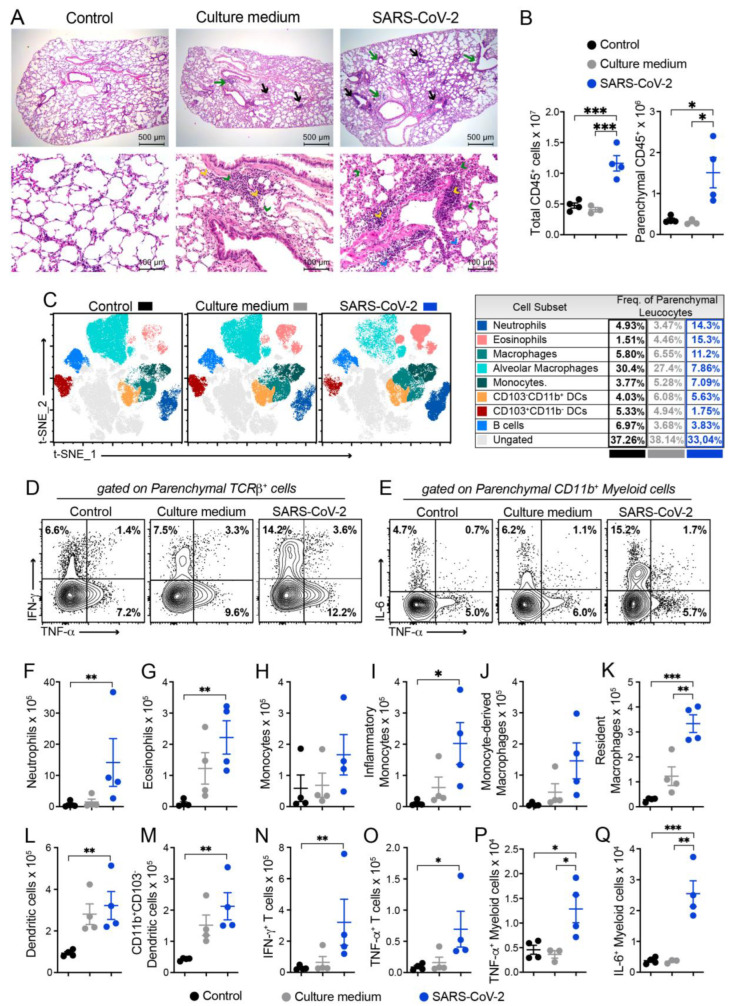
Characterization of the inflammatory process induced by IN instillation of inactivated SARS-CoV-2 by histopathological and cytometry analyses. C57BL/6 mice were instilled with the virus (3 doses of 4 × 10^5^ PFU/each) on days 1, 3, and 5. On the 7th day, the upper left lobe and whole right lung were collected for histopathological and cytometric analyses, respectively. For histopathological evaluation, the samples were washed, fixed, stained with H&E, and then evaluated concerning the presence of inflammatory foci. Then, 5 um thick sections from the control (saline), culture medium, and SARS-CoV-2 groups were analyzed, and the representative images are shown in (**A**). Inflammatory foci around the vessels (black arrows) and around the bronchi (green arrows), neutrophilic infiltrates (green arrow head), macrophage infiltrates (blue arrow head), and lympho-cystic infiltrates (yellow arrow head). The cells were isolated from the lung tissue and total CD45^+^ leukocytes or the parenchymal infiltrating leukocyte fraction (identified based on anti-CD45 intravenous injection) were quantified by flow cytometry (**B**) and specific cells subsets were evaluated according to the gating strategy described in Appendix A. t-distributed stochastic neighbor embedding (t-SNE) analysis illustrating the distribution of cell clusters in each experimental group (**C**) according to the gate strategy described in Appendix A. The table on the right side indicates the frequency of each cell cluster relative to the CD45^+^ parenchyma-infiltrating leukocyte in the control (black), culture medium (grey), and SARS-CoV-2 (blue) groups. The representative contour plots of IL-6, TNF-α, or IFN-γ staining in parenchymal TCRβ^+^ T cells (**D**) or CD11b^+^ myeloid cells (**E**), according to gate strategy described in Appendix A. The total cell numbers of each parenchymal-infiltrating cell subsets are expressed in mean ± SEM, including neutrophils (**F**), eosinophils (**G**), monocytes (**H**), inflammatory monocytes (**I**), monocyte-derived macrophages (**J**), resident macrophages (**K**), dendritic cells (**L**), CD11b^+^CD103^-^dendritic cells (**M**), IFN-γ^+^ TCRβ^+^ cells (**N**), TNF-α^+^ TCRβ^+^ cells (**O**), TNF-α^+^ CD11b+ myeloid cells (**P**), and IL-6^+^CD11b^+^ myeloid cells (**Q**). Data are derived from one experiment between two showing similar findings (n = 4). The results are presented in median and interquartile intervals and the comparison between the groups was performed using the Kruskal–Wallis test, followed by Dunn’s test. * *p* < 0.05; ** *p* < 0.01; *** *p* < 0.001.

**Figure 3 cells-12-01092-f003:**
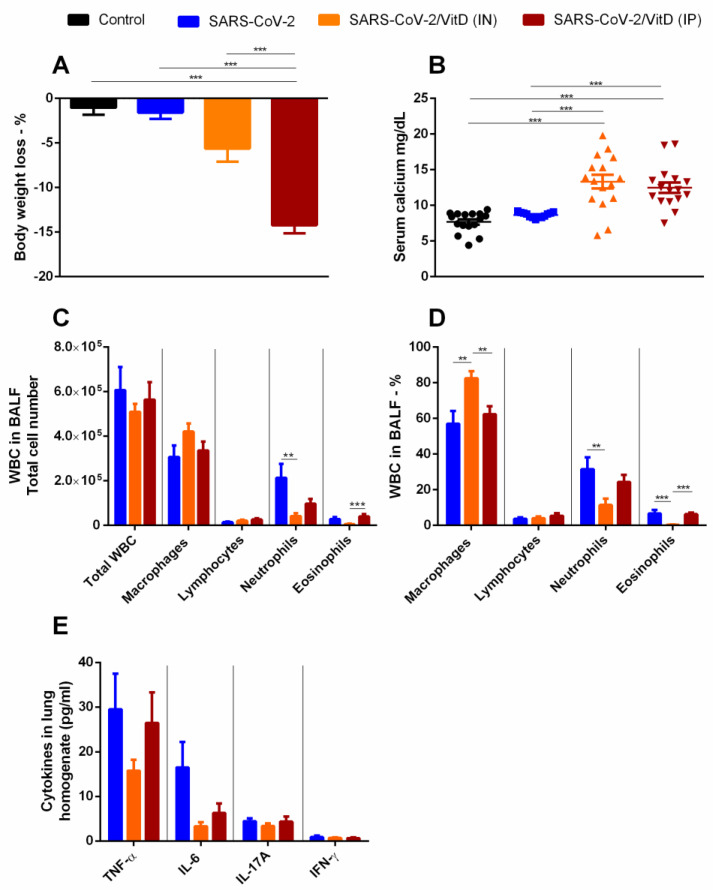
VitD effects on body weight, serum calcium levels, BALF cell counts, and cytokine levels in lung homogenates form mice of mice instilled with inactivated SARS-CoV-2. C57BL/6 mice were instilled with the virus (3 doses of 4 × 10^5^ PFU/each) on days 1, 3, and 5. Body weight was checked daily and, at the 7th day, we collected blood samples for calcium measurement, BALF for WBCs analysis, and lower left lobe for cytokine quantification. Body weight loss (**A**), serum calcium levels (**B**), total WBCs, and WBCs subsets in BALF (**C**), percentage of WBCs in BALF (**D**), and cytokines in lung homogenates (**E**). Data shown in (**A**,**B**,**D**) derive from 3 experiments which were combined (n = 5 mice/experimental group). Data shown in (**A**,**C**–**E**) derive from 2 experiments with similar results which were combined (n = 16 mice/experimental group). The results are expressed as mean ± SEM and the comparison between the groups was performed using an ANOVA followed by Tukey’s test. The results shown in (**B**) are expressed as median and interquartile intervals and the comparison between the groups was performed by the Kruskal–Wallis test followed by Dunn’s test. ** *p* < 0.01; *** *p* < 0.001.

**Figure 4 cells-12-01092-f004:**
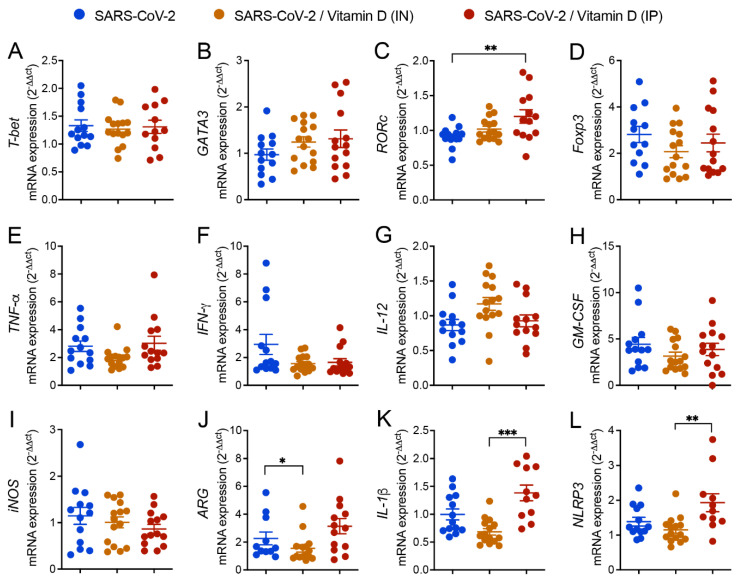
Effect of IN and IP VitD delivery on T cell transcription factors, cytokines, and inflammasome gene transcripts in lungs of mice IN challenged with UV-inactivated SARS-CoV-2. C57BL/6 mice were instilled with the virus (3 doses of 4 × 10^5^ PFU/each) on days 1, 3, and 5. In the IN protocol, mice were treated with 3 VitD doses (0.1 µg/dose) simultaneously with the SARS-CoV-2 inoculum. In the IP protocol, each animal was treated with 4 VitD doses delivered on days 0, 2, 4, and 6. On the 7th day, the lower left lobe was removed, and the RNA extracted and submitted to RT-PCR. Tested genes included *T*-*bet* (**A**), *GATA3* (**B**), *RORc* (**C**), *Foxp3* (**D**), *TNF*-*α* (**E**), *IFN*-*γ* (**F**), *IL*-*12* (**G**), *GM*-*CSF* (**H**), *iNOS* (**I**), *ARG* (**J**), *IL*-*1β* (**K**), *NLRP3* (**L**). Data derive from three experiments with similar results which were combined (n = 11–15 mice/experimental group). The results are presented in median and interquartile intervals and the comparison between the groups was performed by the Kruskal–Wallis test followed by Dunn’s test. * *p* < 0.05; ** *p* < 0.01; *** *p* < 0.001.

**Figure 5 cells-12-01092-f005:**
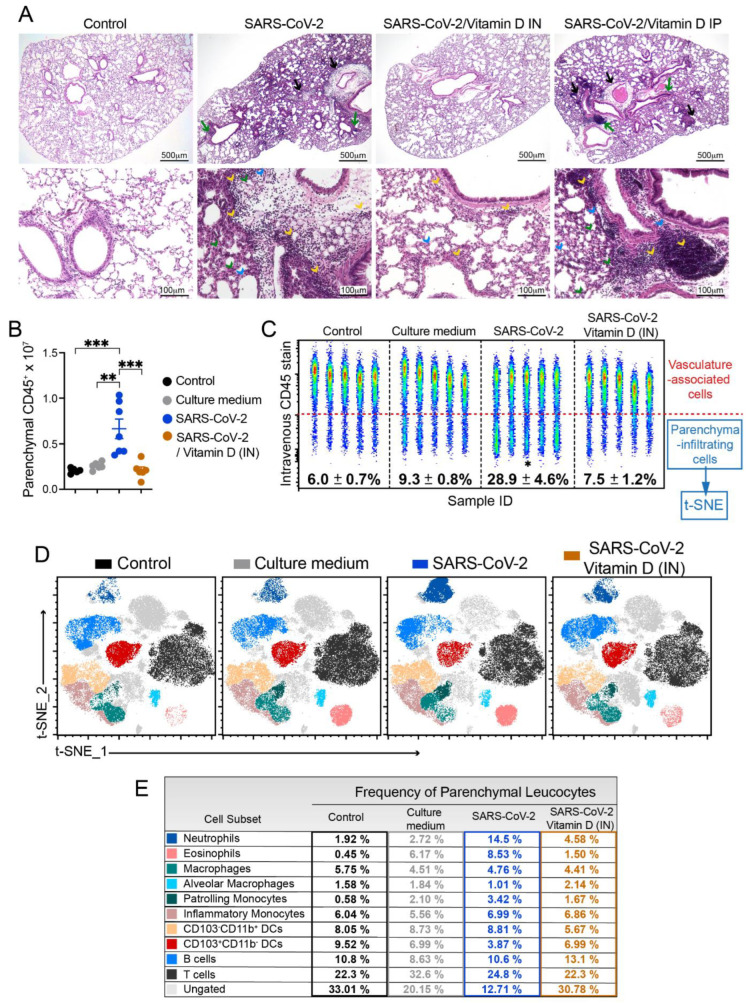
Effect of IN and IP VitD on lung histopathology (**A**) and effect of IN VitD on lung infiltration of pro-inflammatory cells (**B**–**E**) triggered by IN instillation of UV-inactivated SARS-CoV-2. C57BL/6 mice were instilled with the virus (3 doses of 4 × 10^5^ PFU/each) on days 1, 3, and 5. In the IN protocol, mice were treated with 3 VitD doses (0.1 µg/dose) simultaneously with the SARS-CoV-2 inoculum. In the IP protocol, each animal was treated with 4 VitD doses delivered on days 0, 2, 4, and 6. On the 7th day, the upper left lobe and the right lung were collected for histopathological and flow cytometry analyses, respectively. The upper left lobe was washed, fixed, and stained with H&E, and then evaluated concerning the presence of inflammatory foci (**A**) around the vessels (black arrows) and around the bronchi (green arrows), neutrophilic infiltrates (green arrow head), macrophage infiltrates (blue arrow head), and lympho-cystic infiltrates (yellow arrow head). Cells from lung parenchyma were eluted and analyzed after labeling with an array of specific antibodies (**B**–**E**). (**B**) Total numbers of CD45^+^ parenchymal infiltrating leukocyte fraction (identified based on anti-CD45 intravenous injection). (**C**) Representative dot plot of 5 concatenated samples from all groups illustrating the average and SEM of % CD45-negative cells (parenchymal fraction) in lungs from all experimental groups. The specific cell subsets quantified by flow cytometry were evaluated according to the gating strategy described in Appendix A. (**D**) t-distributed stochastic neighbor embedding (t-SNE) analysis illustrating the distribution of cell clusters in each experimental group according to gate strategy described in the Appendix A. (**E**) Table indicating the frequency of each cell cluster relative to the CD45^+^ parenchyma-infiltrating leukocytes in the control (black), culture medium (grey), SARS-CoV-2 (blue) groups and SARS-CoV-2 IN VitD-treated group (orange). Data shown in A are derived from one experiment (n = 5 animals/experimental group) and data shown in (**B**–**E**) are derived from one experiment (n = 7–8 animals/group). Results are presented in median and interquartile intervals and the comparison between the groups was performed using the Kruskal–Wallis test followed by the Dunn’s test. ** *p* < 0.01; *** *p* < 0.001.

**Figure 6 cells-12-01092-f006:**
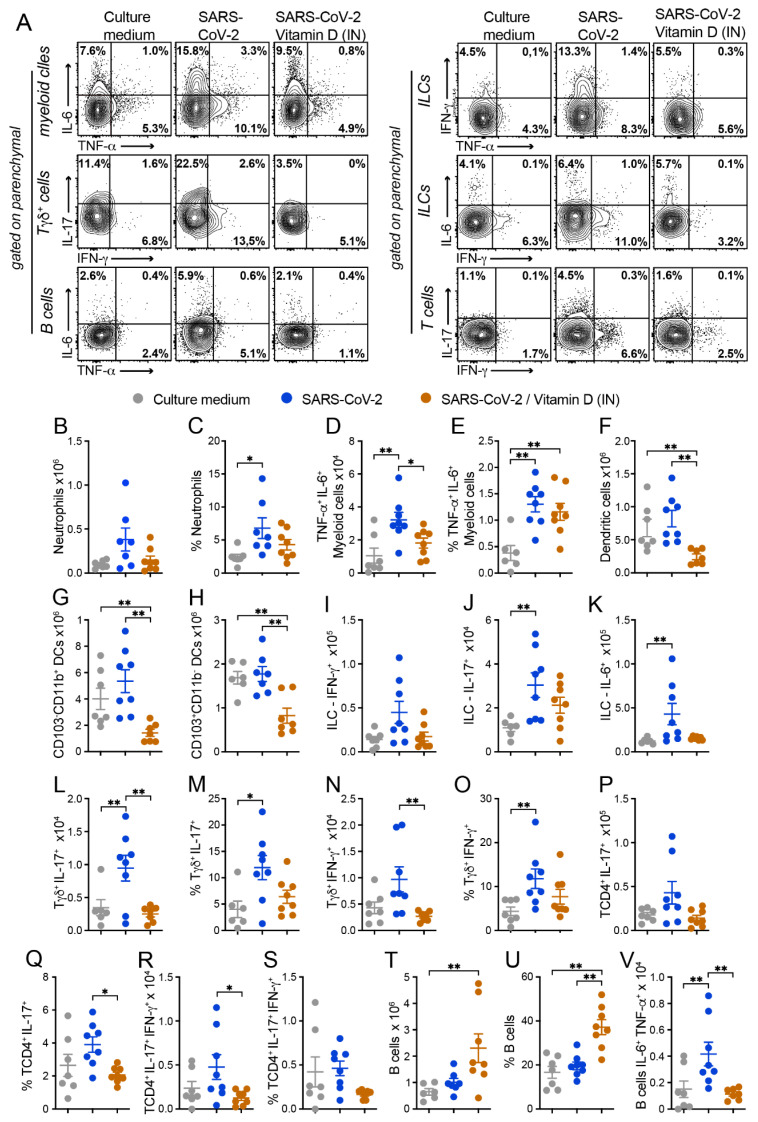
Effect of IN VitD on cell infiltration triggered by IN instillation of UV-inactivated SARS-CoV-2. C57BL/6 mice were instilled with the virus (3 doses of 4 × 10^5^ PFU/each) on days 1, 3, and 5. Mice were treated with 3 VitD doses (0.1 µg/dose) simultaneously with the SARS-CoV-2 inoculum. On the 7th day, the upper left lobe and the right lung were collected for flow cytometry analyses. In order to differentiate the parenchyma-infiltrating leukocytes from the vasculature-associated fraction, mice were intravenously injected with FITC-labeled anti-CD45 antibody 3 min before euthanasia. Specific cell subsets infiltrating the lungs of mice were analyzed according to the gate strategy described in Appendix A. Intracellular cytokine production was detected by flow cytometry in PMA/Ionomycin/brefeldin in vitro stimulated cells. (**A**) Representative contour plots of each experimental group indicating the frequency of cytokine production by each parenchymal cell subset, as indicated in the *y* axis of the figure. (**B**–**V**) Absolute numbers and/or percentage of each cell subset or cytokine-producing cell as indicated in each graph *y* axis. Data derive from one experiment (n = 7–8 animals/group), results are presented in median and interquartile intervals, and the comparison between the groups was performed using the Kruskal–Wallis test followed by Dunn’s test. * *p* < 0.05 ** *p* < 0.01.

## Data Availability

Not applicable.

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
