# Peer review of "Lung Inflammation Induced by Inactivated SARS-CoV-2 in C57BL/6 Female Mice Is Controlled by Intranasal Instillation of Vitamin D"

_cells, 2023, doi:10.3390/cells12071092_

Round 1

Reviewer 1 Report

The manuscript entitled "Experimental lung inflammation induced by inactivated SARS-CoV-2 is controlled by intranasal instillation of vitamin D" reports important findings about production of an experimental inflammation in lungs of mice using inactivated SARS-CoV-2, and controlling this inflammation using VitD. The study is well-designed and the manuscript was prepared well. However, it should be noted that the experiments were performed using inactivated virus. Therefore, the validity of results in case of active disease (live virus) is unknown, which was indicated as a limitation of the study in the manuscript.   

Author Response

Revisor # 1

O manuscrito intitulado "Inflamação pulmonar experimental induzida por SARS-CoV-2 inativado é controlado por instilação intranasal de vitamina D" relata importantes descobertas sobre a produção de uma inflamação experimental em pulmões de camundongos usando SARS-CoV-2 inativado e controlando essa inflamação usando VitD. O estudo está bem desenhado e o manuscrito foi bem preparado. No entanto, deve-se notar que os experimentos foram realizados usando um vírus inativado. Portanto, a validade dos resultados em caso de doença ativa (vírus vivo) é desconhecida, o que foi apontado como uma limitação do estudo no manuscrito.  

Agradecimentos: Agradecemos imensamente o tempo e a atenção dispensados ​​ao nosso trabalho. Seu comentário sobre a validade do nosso modelo é absolutamente pertinente. Dados derivados de um estudo em andamento em nosso grupo de pesquisa mais informações derivadas da literatura foram inseridos na seção de discussão para apoiar a relevância de nosso modelo. Sua contribuição nos permitiu apresentar um manuscrito mais claro e possivelmente mais interessante. Por favor, encontre em anexo uma nova versão do manuscrito com mudanças de trilha destacadas .

 De fato, concordamos que os resultados não nos permitem concluir que o vitD controlaria a inflamação durante uma infecção ativa por SARS-CoV-2. Embora a vitD tenha a capacidade de ativar a resposta imune inata por meio da indução de peptídeos antimicrobianos, isso provavelmente não será suficiente para controlar a multiplicação do vírus e, portanto, o processo inflamatório. Gostaríamos de esclarecer que nosso objetivo neste estudo foi estabelecer um modelo inflamatório pulmonar semelhante ao desencadeado durante a infecção por SARS-CoV-2 em camundongos, mas induzido pelo vírus inativado e também avaliar a capacidade da vitD de controlar esse local processo inflamatório. Esses dois objetivos estão agora claramente definidos na seção de introdução. 

Os textos abaixo transcritos foram inseridos na nova versão:

Nesse contexto, o primeiro objetivo de nossa investigação foi caracterizar o processo inflamatório pulmonar induzido pela instilação intranasal de SARS-CoV-2 inativado. (Linhas 85-87)

 Nesse contexto, nosso segundo objetivo foi investigar se o processo inflamatório pulmonar induzido por SARS-CoV-2 inativado poderia ser modulado negativamente pela VitD administrada por ambas as vias, intraperitonial (IP) e intranasal (IN). (Linhas 108-110)

O estudo está bem desenhado e o manuscrito foi bem preparado. No entanto, deve-se notar que os experimentos foram realizados usando um vírus inativado. Portanto, a validade dos resultados em caso de doença ativa (vírus vivo) é desconhecida, o que foi apontado como uma limitação do estudo no manuscrito.  

Responder

A validação de nosso modelo como uma ferramenta adequada para investigar outros procedimentos para controlar a inflamação pulmonar é corroborada por outras investigações em andamento em nosso grupo de pesquisa. As alterações histológicas que encontramos após a instilação do SARS-CoV-2 inativado por UV são comparáveis ​​à inflamação pulmonar que os camundongos h-ACE2 desenvolvem após a infecção ativa. O perfil de células inflamatórias eluídas do parênquima pulmonar também é muito semelhante ao descrito em nossa investigação (Aype et al, dados não publicados). Esta comparação foi incluída na discussão deste manuscrito; no entanto, os resultados não serão incorporados a este manuscrito porque pertencem a outro estudo em andamento por nosso grupo de pesquisa. 

Veja abaixo as ilustrações gráficas referentes a esses dados:

Figura 1: Infiltrado de células pulmonares após infecção por SARS-CoV-2. Camundongos fêmeas B6.Cg-Tg(K18-ACE2)2Prlmn/J com oito semanas de idade foram inoculados por via intranasal com 50 uL de meio DMEM com 2% de FBS contendo 1x10 5 TCID50 (dose infecciosa de cultura de tecidos) de cultura de tecidos SARS-CoV- 2 ) [SARS-CoV-2 - Brasil /SPBR1/2020, cedido pelo Laboratório de Virologia Clínica e Molecular do ICB-USP. Sete dias após a infecção, os pulmões foram coletados e processados ​​para análise por citometria de fluxo do infiltrado celular.

Figura 2: Produção de citocinas por células pulmonares após infecção por SARS-CoV-2 . Camundongos fêmeas B6.Cg-Tg(K18-ACE2)2Prlmn/J com oito semanas de idade foram inoculados por via intranasal com 50 uL de meio DMEM com 2% de FBS contendo 1x10 5 TCID50 (dose infecciosa de cultura de tecidos) de cultura de tecidos SARS-CoV- 2 ) [SARS-CoV-2 - Brasil /SPBR1/2020, cedido pelo Laboratório de Virologia Clínica e Molecular do ICB-USP. Sete dias após a infecção, os pulmões foram coletados e processados ​​e as células isoladas foram estimuladas in vitro com PMA/Ionomicina para detecção de citocinas por análise de citometria de fluxo do infiltrado celular.

Essas alterações histológicas que encontramos após a instilação do SARS-CoV-2 inativado por UV são comparáveis ​​à inflamação pulmonar que os camundongos desenvolvem após a infecção ativa (Aype et al, dados não publicados). Quando avaliamos o infiltrado celular no pulmão de camundongos infectados, encontramos um aumento significativo no número de células CD45+ no parênquima pulmonar em comparação ao grupo não infectado. Em particular, o aumento da infiltração de leucócitos no parênquima pulmonar refletiu um aumento de neutrófilos, macrófagos e células T (como representado nas figuras acima). Digno de nota, quando as células pulmonares foram estimuladas para detecção de citocinas intracelulares, encontramos um aumento significativo na produção de IFN-gama e TNF-alfa pelas células T e também de TNF-alfa e IL-6 pelas células mieloides CD11b+.

Veja abaixo o texto que foi inserido na seção de discussão informando essas e outras descobertas da literatura para apoiar a utilidade de nosso modelo:

A validação de nosso modelo como uma ferramenta adequada para investigar outros procedimentos para controlar a inflamação pulmonar é corroborada por outra investigação em andamento em nosso grupo de pesquisa. As alterações histológicas que encontramos após a instilação do SARS-CoV-2 inativado por UV são comparáveis ​​à inflamação pulmonar que os camundongos h-ACE2 desenvolvem após a infecção ativa. O perfil de células inflamatórias eluídas do parênquima pulmonar é também muito semelhante ao descrito na nossa investigação (Aype et al, dados não publicados). Esta validação é também reforçada pelos dados descritos por Bi et al., 2021. Estes autores desenvolveram um modelo de síndrome do desconforto respiratório agudo induzido por SARS-CoV-2 por instilação intratraqueal de SARS-CoV-2 inativado por formaldeído. Suas alterações histopatológicas descritas e o perfil de células infiltradas nos pulmões também são semelhantes aos nossos achados. (Linhas 660-669)

Revisor # 1

O manuscrito intitulado "Inflamação pulmonar experimental induzida por SARS-CoV-2 inativado é controlado por instilação intranasal de vitamina D" relata importantes descobertas sobre a produção de uma inflamação experimental em pulmões de camundongos usando SARS-CoV-2 inativado e controlando essa inflamação usando VitD. O estudo está bem desenhado e o manuscrito foi bem preparado. No entanto, deve-se notar que os experimentos foram realizados usando um vírus inativado. Portanto, a validade dos resultados em caso de doença ativa (vírus vivo) é desconhecida, o que foi apontado como uma limitação do estudo no manuscrito.  

Agradecimentos: Agradecemos imensamente o tempo e a atenção dispensados ​​ao nosso trabalho. Seu comentário sobre a validade do nosso modelo é absolutamente pertinente. Dados derivados de um estudo em andamento em nosso grupo de pesquisa mais informações derivadas da literatura foram inseridos na seção de discussão para apoiar a relevância de nosso modelo. Sua contribuição nos permitiu apresentar um manuscrito mais claro e possivelmente mais interessante. Por favor, encontre em anexo uma nova versão do manuscrito com mudanças de trilha destacadas .

 De fato, concordamos que os resultados não nos permitem concluir que o vitD controlaria a inflamação durante uma infecção ativa por SARS-CoV-2. Embora a vitD tenha a capacidade de ativar a resposta imune inata por meio da indução de peptídeos antimicrobianos, isso provavelmente não será suficiente para controlar a multiplicação do vírus e, portanto, o processo inflamatório. Gostaríamos de esclarecer que nosso objetivo neste estudo foi estabelecer um modelo inflamatório pulmonar semelhante ao desencadeado durante a infecção por SARS-CoV-2 em camundongos, mas induzido pelo vírus inativado e também avaliar a capacidade da vitD de controlar esse local processo inflamatório. Esses dois objetivos estão agora claramente definidos na seção de introdução. 

Os textos abaixo transcritos foram inseridos na nova versão:

Nesse contexto, o primeiro objetivo de nossa investigação foi caracterizar o processo inflamatório pulmonar induzido pela instilação intranasal de SARS-CoV-2 inativado. (Linhas 85-87)

 Nesse contexto, nosso segundo objetivo foi investigar se o processo inflamatório pulmonar induzido por SARS-CoV-2 inativado poderia ser modulado negativamente pela VitD administrada por ambas as vias, intraperitonial (IP) e intranasal (IN). (Linhas 108-110)

O estudo está bem desenhado e o manuscrito foi bem preparado. No entanto, deve-se notar que os experimentos foram realizados usando um vírus inativado. Portanto, a validade dos resultados em caso de doença ativa (vírus vivo) é desconhecida, o que foi apontado como uma limitação do estudo no manuscrito.  

Responder

A validação de nosso modelo como uma ferramenta adequada para investigar outros procedimentos para controlar a inflamação pulmonar é corroborada por outras investigações em andamento em nosso grupo de pesquisa. As alterações histológicas que encontramos após a instilação do SARS-CoV-2 inativado por UV são comparáveis ​​à inflamação pulmonar que os camundongos h-ACE2 desenvolvem após a infecção ativa. O perfil de células inflamatórias eluídas do parênquima pulmonar também é muito semelhante ao descrito em nossa investigação (Aype et al, dados não publicados). Esta comparação foi incluída na discussão deste manuscrito; no entanto, os resultados não serão incorporados a este manuscrito porque pertencem a outro estudo em andamento por nosso grupo de pesquisa. 

Veja abaixo as ilustrações gráficas referentes a esses dados:

Figura 1: Infiltrado de células pulmonares após infecção por SARS-CoV-2. Camundongos fêmeas B6.Cg-Tg(K18-ACE2)2Prlmn/J com oito semanas de idade foram inoculados por via intranasal com 50 uL de meio DMEM com 2% de FBS contendo 1x10 5 TCID50 (dose infecciosa de cultura de tecidos) de cultura de tecidos SARS-CoV- 2 ) [SARS-CoV-2 - Brasil /SPBR1/2020, cedido pelo Laboratório de Virologia Clínica e Molecular do ICB-USP. Sete dias após a infecção, os pulmões foram coletados e processados ​​para análise por citometria de fluxo do infiltrado celular.

Figura 2: Produção de citocinas por células pulmonares após infecção por SARS-CoV-2 . Camundongos fêmeas B6.Cg-Tg(K18-ACE2)2Prlmn/J com oito semanas de idade foram inoculados por via intranasal com 50 uL de meio DMEM com 2% de FBS contendo 1x10 5 TCID50 (dose infecciosa de cultura de tecidos) de cultura de tecidos SARS-CoV- 2 ) [SARS-CoV-2 - Brasil /SPBR1/2020, cedido pelo Laboratório de Virologia Clínica e Molecular do ICB-USP. Sete dias após a infecção, os pulmões foram coletados e processados ​​e as células isoladas foram estimuladas in vitro com PMA/Ionomicina para detecção de citocinas por análise de citometria de fluxo do infiltrado celular.

Essas alterações histológicas que encontramos após a instilação do SARS-CoV-2 inativado por UV são comparáveis ​​à inflamação pulmonar que os camundongos desenvolvem após a infecção ativa (Aype et al, dados não publicados). Quando avaliamos o infiltrado celular no pulmão de camundongos infectados, encontramos um aumento significativo no número de células CD45+ no parênquima pulmonar em comparação ao grupo não infectado. Em particular, o aumento da infiltração de leucócitos no parênquima pulmonar refletiu um aumento de neutrófilos, macrófagos e células T (como representado nas figuras acima). Digno de nota, quando as células pulmonares foram estimuladas para detecção de citocinas intracelulares, encontramos um aumento significativo na produção de IFN-gama e TNF-alfa pelas células T e também de TNF-alfa e IL-6 pelas células mieloides CD11b+.

Veja abaixo o texto que foi inserido na seção de discussão informando essas e outras descobertas da literatura para apoiar a utilidade de nosso modelo:

A validação de nosso modelo como uma ferramenta adequada para investigar outros procedimentos para controlar a inflamação pulmonar é corroborada por outra investigação em andamento em nosso grupo de pesquisa. As alterações histológicas que encontramos após a instilação do SARS-CoV-2 inativado por UV são comparáveis ​​à inflamação pulmonar que os camundongos h-ACE2 desenvolvem após a infecção ativa. O perfil de células inflamatórias eluídas do parênquima pulmonar é também muito semelhante ao descrito na nossa investigação (Aype et al, dados não publicados). Esta validação é também reforçada pelos dados descritos por Bi et al., 2021. Estes autores desenvolveram um modelo de síndrome do desconforto respiratório agudo induzido por SARS-CoV-2 por instilação intratraqueal de SARS-CoV-2 inativado por formaldeído. Suas alterações histopatológicas descritas e o perfil de células infiltradas nos pulmões também são semelhantes aos nossos achados. (Linhas 660-669)

Reviewer 2 Report

Please, find my comments in the attached file.

Author Response

Reviewer # 2

In this paper, the Authors examined the effects of intranasal instillation of Vitamin D3 in C57BL/6 mice where an inflammatory COVID-19-like pulmonary process was developed through the IN instillation of UV-inactivated SARS-CoV-2. They analyzed the variation of type and number of cells in the BALF and the lung parenchyma, performed the histopathological analysis, and quantified T cell subsets, inflammatory mediators, and several cytokines. In my opinion, this paper describes a relevant amount of work which, however, suffers from some important weaknesses that hinder its acceptance in its present form. Please, find below a non-exhaustive list of observations.

Answer

Acknowledgment: We are very thankful to the reviewer for the exceptional  time and attention given to our work. The questions, suggestions and criticisms were all welcome because they were relevant and allowed us to submit a more clear and possibly more interesting manuscript. Please find enclosed a new version of the manuscript with highlighted track changes.

  • The scientific rationale is not fully explained. Considering the number of papers reporting the beneficial effect of vitD administration by different routes in different inflammatory experimental designs, the Authors should explain why they chose intranasal administration in an experimental model mimicking a severe syndrome primarily involving pulmonary parenchyma.

Answer

To the best of our knowledge, most of these trials were done by oral administration of VitD, which is, considering some limitations, a route that allows a systemic drug distribution (Lou et al., 2023).  In this context, our second objective was to investigate if the lung inflammatory process induced by inactivated SARS-CoV-2 could be downmodulated by VitD administered by both, intraperitoneal (IP) and intranasal (IN) routes. The choice of the IP route was based on our previous experience showing that vitD by this via was able to control the central nervous system CNS) inflammation in an experimental murine model of multiple sclerosis (de Oliveira et al., 2020).The decision to test vitD by IN route was adopted considering different reasons. Initially we thought about practical issues as, for example, non-invasiveness, possible immediate effect considering that vitD would be applied directly at the inflammatory site and even the possibility of self-administration. We also considered the fact that previous reports indicate that vitD has a remarkable anti-inflammatory effect when locally applied at the respiratory system. This has already been demonstrated in some lung experimental conditions such as rhinitis (Cho et al., 2029) and asthma (Feng et al., 2021). In addition, the in situ application of vitD has also been effective in other localized pro-inflammatory diseases as for example vitiligo (Forschner et al., 2007) and psoriasis (Kieffer, 2004). The fact that IN vitD could theoretically control, at least partially, some of the immediate or late neurological alterations caused by dissemination of SARS-CoV-2 to the nervous system, was also pondered. This possibility was based on reports showing that vit D attenuates blood-brain barrier disruption (Enkhjard et al, 2017), therefore decreasing the entry of inflammatory cells into the central nervous system. In addition, the nose-to-brain route has been proposed as a promising strategy for drug delivery to the brain (Wang et al., 2019).

This rationale was included in the introduction section (Lines 106-127)

  • Why the Authors choose only female mice? Why they did not balance m/f mice, to obtain experimental results not specifically linked to one sex?

Answer

This choice was made in the context of our main research area that is the effect of immunomodulators, including vitD, on the central nervous system of the murine experimental autoimmune encephalomyelitis (EAE) which is an animal model of multiple sclerosis. As female C57BL/6 mice are twice as likely to develop MS than males  and all our previous experience of vitD effect on EAE is with females, we choose this gender. In this context, our initial  proposal was to evaluate  the lung inflammatory process induced by the inactivated and the active virus (other ongoing study by our group) to decide if we could use the lung inflammation model induced by the inactivated virus to study the immunomodulatory ability of vitD to control this process.  We also planned to investigate the effect of lung  inflammation associated to SARS-CoV-2 on EAE evolution considering that T cell-licensing before brain access is possibly occurring in the lung.

This information was not included in the manuscript.

  • Why the Authors showed the VitD effects on body weight and serum calcium levels? Although their variations are commented on in the discussion, their relationship with the subject matter of the paper is not understood.

Answer

These parameters are frequently used to indicate vitamin D intoxication. Some considerations regarding these findings  were included in the discussion section and are transcribed below:

Even though the IP administration of vitD triggered a few down-modulatory effects, this procedure was not able to control lung inflammation. Conversely, this protocol increased IL-1b, NLRP3 and RORc expression, suggesting a possible toxic proinflammatory activity associated with an excess of vitD. Actually, some authors raised the possibility that vitD excess could trigger inflammation through T cell stimulation via hypercalcemia. In this sense, serum calcium levels and body weight loss have been frequently employed to indicate VitD toxicity (DeLuca et al., 2010, Hausler et al., 2020). In healthy individuals, exogenous vitD toxicity is generally associated with the continuous use of high vitD doses (Marcinovska-Suchowierska et al., 2018). Even though only a few vitD doses were employed in our protocols, calcium levels were similarly altered in IP and IN treated mice, possibly excluding the extracellular hypercalcemia in IP vitD-treated mice as the cause of inflammasome activation (Rossol et al., 2012). Of note, the IP vitD-treated animals also lost significantly more weight than the ones treated by IN vitD. If this accentuated body weight loss, which is also an indicative of vitD toxicity, is somehow related to IP  vitD ineffectiveness in controlling lung inflammation is not known yet.As body weight loss during vitD treatment has been attributed to its effect in the brain [63], a simple explanation to the finding that vitD IP causes much more weight loss than IN vitD is that IP vitD determines higher concentration of this vitamin in the brain.  A pharmacodynamic study of the tissue distribution of vitD administered by these 2 routes, especially in the CNS and in the lungs, will be necessary to understand this differential effect. (Lines 752-770)

  • The number of experiments is not always sufficient to justify the statistical processing of the results. This is the case for Fig. 1, 2, 5.

Answer

The number of repetitions of each experiment and the number of  animals per group are informed in each legend. For the sake of clarity, this information is also transcribed below for all figures:

Figure 1

Data shown in A, B and C-P is derived from two experiments with similar results which were combined (n=9), except Tbet (n=8), IL-17 (n=6) and IL-6 (n=4).

Figure 2

Data shown  is derived from one experiment between two showing similar findings (n=4).

Figures 3 A-D: data shown is derived from three independent experiments  which were combined

Figure 3A: 15-16 animals/group

Figure 3B: X-Y animals /group

Figures 3C-D: x-y animals /group

Figure 3E: 2 experiments combined: X-Y animals/group

Figure 4:Data derived from three experiments with similar results which were combined (n=11-15 mice/ group).

Figures 5 and 6: The cytometry analysis in animals treated with vitD was performed only once,  but in this case each experimental group was constituted by 7-8 animals.

  • Fig. 2A The histopathological evaluation, comes from only one experiment, which is not sufficient to draw the conclusions the Authors claim in the paper.

Answer

Figure 2A is representative of 2 experiments whose results were similar (394-395).

  • Fig. 3: The Authors stated: Data shown in A, B and D derive from one experiment (n=5 mice/experimental group). Data showed in C and D derive from 3 experiments with similar results which were combined (n= 16 mice/experimental group). So: data shown in D derive from one or three experiments.

Answer

All data previously shown in figure 3 was reanalyzed because new samples of serum and lung homogenates were used to measure calcium and cytokines, respectively.The number of experiments and samples has been updated as informed in the corresponding legend and transcribed below:

Figure 3A: 3 experiments combined, 15-16 animals/group

Figure 3B: 3 experiments combined, 15-16 animals /group

Figures 3C and 3D: 3 experiments combined, 15-16 animals/group

Figure E: 2 experiments combined, 10 animals/group

  • Fig. 6: The number of experiments is not cited.(567-569)

Answer

This experiment was the only one done once but included a higher number of animals, 8 in each experimental group. This information is now included in the legend.

Some important discrepancies between figures and their description are present.

Page 6, lines 274-276 and Fig. 1a and 1b: Line 274-275: when comparing the total number and the percentages of WBC cells, the decrease of macrophages is significant only if its percentage value compared to the WBC in the BALF is considered, while the increase of lymphocytes is significant only if one considers their number about the total of the WBC and not its percentage value. The Authors should explain this.

Answer

Within an heterogeneous cell population, as the one that constitutes the WBC recovered from the BALF, the total number and the percentage of each one will not be, necessarily, similarly affected by a therapeutic procedure. In this case, for example, even though the total amount of macrophages has been slightly higher in the SARS-CoV-2 group than in the two control groups, the percentage of these cells in this group was significantly lower in comparison to the control groups, suggesting that there is an increased number of other cell types in this group (neutrophils in this particular case). Consequently, the percentage of macrophages is lower in this group. The same kind of reasoning applies to the lymphocyte population; in spite of their significantly higher number in SARS-CoV-2 group, their increase in terms of percentage was not statistically significant. We understand that the total number of WBC along with their different cell types, give a clear and more direct idea of the inflammatory extension. On the other hand, the calculation of the percentage of each cell type gives an idea of the pattern of immune response recruited to this compartment. For instance, in our experimental model, the immune tone of the airways is shifted towards a neutrophilic inflammatory profile in detriment of a mononuclear or eosinophilic infiltrate. Therefore, we found an increase in both percentage and total counts of neutrophils, but not in the other cell subsets.

The explanation concerning the meaning of these two parameters (total number and percentage of each subset) was slightly modified and inserted at the discussion section as transcribed below::

Altogether,  the total number of WBC along with their different cell types, gave a clear and more direct idea of the inflammatory extension. On the other hand, the calculation of the percentage of each cell type gave an idea of the pattern of immune response recruited to this compartment. In our experimental model, the immune tone of the airways was shifted towards a neutrophilic inflammatory profile in detriment of a mononuclear or eosinophilic infiltrate. Therefore, we found an increase in both percentage and total counts of neutrophils, but not in the other cell subsets. (Lines 585-592)

Page. 8, lines 338-339 and Figure 2H, 2J: The Authors claim that “The predominant profile of all tested cells was characterized by a significantly higher number of cells enriched in the t-SNE analysis in the SARS-CoV-2 group in comparison to the control group”, but graphs showing the total number of monocytes (2H) and monocytes derived macrophages (2J) lacks this claimed statistical significance.

Answer

When we run the t-SNE, algorithm, these two cell subsets did not clusterize in the same way as the other myeloid cells.

Page 10, lines 384-86: the Authors claimed “BALF findings suggested a modulatory action of VitD administered by both routes and characterized by reduction, although not statistically significant, of lymphocytes, macrophages, neutrophils and eosinophils, as illustrated in Figures 3C and D.” Actually, In Figure 3C and 3D, it can be observed that in WBC in BALF, expressed as total cell number, after IP instillation of VitD there is a significant decrease of neutrophils in relation to control, while in WBC in BALF as a percentage, there is a decrease of macrophages in relation to control, a decrease of lymphocytes in relation to SARS-CoV-2 and an increase of neutrophils in relation to control.

Answer

A larger number of experiments concerning BALF findings (Figures 3C and 3D) and also the other parameters originally included in this figure were reanalyzed and showed more clear differences among the groups.

Concerning the BALF, the comparison among SARS-CoV-2, SARS-CoV-2/vitD (IP) and SARS-CoV-2/vitD (IN) showed no differences in the total number of WBC, macrophages and lymphocytes. However, a significant decrease of PMNs was detected in the SARS-CoV-2/vitD (IN)-treated group in comparison to the non-treated control. Additionally, the number of eosinophils in the SARS-CoV-2/vitD (IP) group was significantly higher than in the IN-treated one. Concerning the percentage of these cells, a higher percent of macrophages was found in the IN-treated group in comparison with the two other groups and  a decreased percentage of PMNs and eosinophils was observed in the IN-treated group in relation to the non-treated one. The percent of eosinophils in the IN-treated mice was also significantly reduced in comparison to the IP-treated ones.). (Lines 406-416)

Page 10, lines 390-392: the Authors claimed: “in Figure 3E, the SARS-CoV-2 group produced higher levels of IL-17A, TNF, and IL-6. The level of these proinflammatory cytokines was reduced by both VitD administration routes, being this effect more accentuated by IN pathway” This statement has no confirmation in the graph shown in Figure 3E, where no variation in the level of cytokines has statistical significance, except TNF (alpha?) which level is higher in SARS-Cov-2 samples.

Answer

A higher number of lung homogenates was tested concerning cytokines concentration. Even though  TNF-a levels in the IN-treated group and IL-6 in IN and IP-treated groups presented a tendency to be lower in comparison to the non-treated group, these differences were discrete. An accurate description of Figure 3E was included at the results section and is transcribed below:

To analyze if SARS-CoV-2-induced lung inflammation model was also mimicking the cytokine storm-like phenomenon and to reinforce the presumed down-modulatory effect of VitD, we tested the presence of pro-inflammatory and regulatory cytokines in lung homogenates. As can be observed in Figure 3E, IN vitD decreased TNF-a and IL-6 and IP vitD decreased IL-6 levels, however, these alterations were not significant. No changes were detected in IL-17A and IFN-g (Figure 3E) or in the other tested cytokines as IL-2, IL-4 and IL-10 (not shown). (Lines 416-422)

Page 12, line 413: the Authors stated: “IN VitD treated-mice displayed a reduction, although not statistically significant, in the expression of TNF-α, IFN-γ, GMF-CSF, ARG, IL-1β and NLRP3 in comparison to the SARS-CoV-2 untreated mice Figures 4E, F, H, J, K and L). The analysis of the graph shows that the decrease of ARG in the INVitD group is statistically significant compared to SARS-Cov-2.

Answer

The wrong information was corrected:

“IN VitD treated-mice displayed a slight reduction in the expression of TNF-α, IFN-γ, GM-CSF, IL-1β and NLRP3 (Figures 4E, F, H, K and L, respectively) and a significantly reduction in ARG (Figure 4J) in comparison to the SARS-CoV-2 untreated mice. (Lines 443-445).

Page 13, line 434-436: the Authors stated: “Even though BALF and cytokines in lung homogenates have suggested a possible down-modulatory effect of IP VitD, this modulatory activity was not confirmed by histopathological analysis of the lungs. Actually, in Figure 4, panels J and K, both NLRP3 and IL1beta increased in a statistically significant way after IP administration of vitamin D to IN administration. This suggests a possible systemic pro-inflammatory effect of vitamin D in attenuated SARS-CoV_2 infected mice.

Answers (A and B)

  1. First of all we would like to clarify that our lung inflammation model was induced by using an UV inactivated SARS-CoV-2 preparation and not an attenuated virus. This information is now clearly stated in the abstract and the introduction. Please see the transcribed informations from both sections below:

Abstract:

Here, we described the induction of a pulmonary inflammatory process triggered by the intranasal (IN) instillation of UV-inactivated SARS-CoV-2 in C57BL/6 mice and then the evaluation of vitamin D (VitD) ability to control this process.( Lines 17-19)

Introduction:

In this context, the first objective of our investigation was to characterize the inflammatory lung process induced by intranasal instillation of inactivated SARS-CoV-2  (Lines 85-87)

In this context, our second objective was to investigate if the lung inflammatory process induced by inactivated SARS-CoV-2 could be downmodulated by VitD administered by both, intraperitoneal (IP) and intranasal (IN) routes. The choice of the IP route was based on our previous experience showing that vitD by this via was able to control the central nervous system CNS) inflammation in an experimental murine model of multiple sclerosis (de Oliveira et al., 2020). (Lines 108-113).

  1. According to the reviewer hypothesis, the elevated expression of IL-1b and NLRP3 (and also RORc, in our opinion) could trigger a systemic proinflammatory state, explaining therefore the ineffectiveness of this route of administration. This possibility makes sense considering that we used 4 vitD doses during the IP protocol and only 3 doses during the IN protocol. Even though the literature  concerning vitD effects over the immune system is mostly directed to its downmodulatory potential, emerging studies alert for a possible pro-inflammatory activity of excessive vitD supplementation triggered by a T cell-stimulating effect via secondary hypercalcemia (Hausler & Weber, 2019). This possibility is interesting and  was  inserted at the discussion section:

Even though the IP administration of vitD triggered a few down-modulatory effects, this procedure was not able to control lung inflammation. Conversely, this protocol increased IL-1b, NLRP3 and RORc expression, suggesting a possible toxic proinflammatory activity associated with an excess of vitD. Actually, some authors raised the possibility that vitD excess could trigger inflammation through T cell stimulation via hypercalcemia. In this sense, serum calcium levels and body weight loss have been frequently employed to indicate VitD toxicity (DeLuca et al., 2010, Hausler et al., 2020). In healthy individuals, exogenous vitD toxicity is generally associated with the continuous use of high vitD doses (Marcinovska-Suchowierska et al., 2018). Even though only a few vitD doses were employed in our protocols, calcium levels were similarly altered in IP and IN treated mice, possibly excluding the extracellular hypercalcemia in IP vitD-treated mice as the cause of inflammasome activation (Rossol et al., 2012). Of note, the IP vitD-treated animals also lost significantly more weight than the ones treated by IN vitD. If this accentuated body weight loss, which is also  indicative of vitD toxicity, is somehow related to IP  vitD ineffectiveness in controlling lung inflammation is not known yet. As body weight loss during vitD treatment has been attributed to its effect in the brain [63], a simple explanation for the finding that vitD IP causes much more weight loss than IN vitD is that IP vitD determines higher concentration of this vitamin in the brain.  A pharmacodynamic study of the tissue distribution of vitD administered by these 2 routes, especially in the CNS and in the lungs, will be necessary to understand this differential effect. (Lines 752-770)

Page 15, lines 488-489: the Authors stated: “but not the frequency of the CD103+CD11b- (Figure 5B and D).” Actually, Fig. 5B refers to Parenchymal CD45+ X 107 cells and 5D to the frequency of parenchymal Leukocytes in Control, Medium, SARS-CoV-2 and SARS-CoV-2 IN-vitD treated. Fig. 5 lacks C, D, and E panels legend.

Answer

Indeed, the figure number is wrong and we replaced (Figure 5B and 5D by Figure 5D and 5E in the manuscript). The legend of Fig 5 was corrected as described  below and inserted in the new manuscript version.

Figure 5. Effect of IN VitD and IP vitD on lung histopathology (A) and effect of IN vitD on  lung infiltration of cytokine-producing cells (B, C, D, E, F-Q) triggered by IN instillation of UV inactivated SARS-CoV-2. C57BL/6 mice were instilled with the virus (3 doses of 4.105 PFU/each) on days 1, 3 and 5. In the IN protocol, mice were treated with 3 VitD doses (0,1 µg/dose) simultaneously with the SARS-CoV-2 inoculum. In the IP protocol, each animal was treated with 4 VitD doses delivered on days 0,2,4 and 6. At the 7th day, the upper left lobe and the right lung were collected for histopathological and flow cytometry analyses, respectively. The upper left lobe was washed, fixed and stained with H&E, and then evaluated concerning the presence of inflammatory focci (A). Cells from lung parenchyma were eluted and analyzed after labeling with an array of specific antibodies (B-E). (B) Total numbers of CD45+ parenchymal infiltrating leukocyte fraction (identified based on anti-CD45 intravenous injection). (C) Representative dot plot of 5 concatenated samples from all groups illustrating the average and SEM of % CD45 negative cells (parenchymal fraction) in lungs from all experimental groups. The specific cell subsets quantified by flow cytometry were evaluated according to the gating strategy described in Supplementary Figure 2. (D) t-distributed stochastic neighbor embedding (t-SNE) analysis illustrating the distribution of cell clusters in each experimental group according to gate strategy described in the Supplementary Figure 2. (E) Table indicating the frequency of each cell cluster relative to the CD45+ parenchyma infiltrating leukocytes in the Control (black), Culture medium (grey), SARS-CoV-2 (blue) groups and SARS-CoV-2 IN ViTD-treated group (orange). Data shown in A is derived from one experiment (n=5 animals/experimental group) and data shown in B-E is derived from one experiment (n= 7-8 animals/group). Results are presented in median and interquartile intervals and the comparison between the groups was performed by the Kruskal-Wallis test followed by the Dunn´s test. * p<0.05; ** p<0.01. (490-511)

A major problem is the number of experiments, which is not sufficient to establish correct statistics.

Answer

More clear information concerning the number of experiments and of animals in each experimental  group is now available in each corresponding legend.  Except the cytometry analysis related to the effect of vitD on lung inflammation (Figures 5 and 6), all the other experiments were done more than once.

  • Minor points: o Page 6, lines 281-282- onwards: it is not clear how to correlate the relative expression of several genes by RT-PCR to the alterations in T cell subsets which are mentioned in the title

Answer

This title (ítem 3.2) was chosen because it is well accepted that the level of expression of certain genes coding for transcription factors as Tbet, GATA3, RORc and Foxp3 are usually correlated with the level of expansion of Th1, Th2, Th17 and Tregs, respectively. For this reason we would like to keep this title.

Page. 8, line 331: “is” instead of “as”.

Answer

Text was corrected, as was replaced by is

Reviewer 3 Report

The manuscript focuses on the apparent advantages of IN administration of VitD for the decrease of a pulmonary inflammatory process induced in mice by inactivated SARS-CoV-2. I have numerous concerns on the model and the final conclusions presented in this study.

1) It is not clearly discussed how much the pulmonary inflammatory process induced in mice by inactivated SARS-CoV-2 (in the present study) resembles to the real pulmonary inflammatory process induced by active SARS-CoV-2 (in reality). What are the pathophysiological similarities and differences between the two processes? In other words, how useful is the present model for conclusions on real life SARS-CoV-2 inflammation?

2) Although in the Conclusion, the study claims that "...most of the clinical trials done with VitD employed other supplementation routes, mainly the oral one, we suppose that the IN one could be more efficient...". However, oral adminsitration was not tried in the present study, only IP. Thus, such conclusion can be drawn only for IP administration.

3) It is also claimed in the Discussion that "Theoretically, a higher local concentration of VitD could be more efficient and, possibly, less toxic than systemic administration. Indeed, the higher efficacy of local delivery was confirmed by our findings..." The study did not provide data on local concentration of VitD. The "efficacy of local delivery" is not defined in pharmacokinetics. I cannot see any data on local delivery.

4) Again the statement (in the Discussion) "Considering that the local delivery efficacy was much more impressive than the systemic, we believe that most of VitD effects are locally occurring and are being mediated by some of its well-established immunomodulatory mechanisms." is not supported by any data and the term "local delivery efficiency" is not defined. What is the unit of local delivery efficiency??? And how did you measure that?

5) To draw relevant conclusions and explain the difference in pharmacokinetics of the different administration routes (IN and IP), the measurement of systemic and local concentrations were necessary. The present manuscript does not provide enough data to explain the apparent success of IN administration and I am not convinced that it may be real alternative of oral adminsitration for VitD.

Author Response

Reviewer # 3 

The manuscript focuses on the apparent advantages of IN administration of VitD for the decrease of a pulmonary inflammatory process induced in mice by inactivated SARS-CoV-2. I have numerous concerns on the model and the final conclusions presented in this study.

Acknowledgment: We are very thankful to the reviewer for the time and attention given to our work. The criticisms were all welcome because they were relevant and allowed us to submit a more clear and possibly more interesting manuscript. Please find enclosed a new version of the manuscript with highlighted track changes

Answer

In fact, this question was also raised by Reviewer #2 and we believe that more clarification of the rationale will allow a better understanding of  the model used in our study.  

1) It is not clearly discussed how much the pulmonary inflammatory process induced in mice by inactivated SARS-CoV-2 (in the present study) resembles  the real pulmonary inflammatory process induced by active SARS-CoV-2 (in reality). What are the pathophysiological similarities and differences between the two processes? In other words, how useful is the present model for conclusions on real life SARS-CoV-2 inflammation?

Answer

            The validation of our model as an adequate tool to investigate other procedures to control lung inflammation is supported by another investigation going on in our research group. The histological changes we found after the instillation of the UV-inactivated SARS-CoV-2 are comparable to the lung inflammation that mice develop after the active infection (Aype et al, unpublished data). When we evaluated the cell infiltrate in the lung of infected mice, we found a significant increase in the numbers of CD45+ cells in the lung parenchyma in comparison to the uninfected group. In particular, the increase in leukocyte infiltration in lung parenchyma reflected an increase in neutrophil, macrophages and T cells.. Of note, when the lung cells were stimulated for intracellular cytokine detection, we found a significant increase in the production of IFN-gamma and TNF-alpha by T cells and also in TNF-alpha and IL-6 by CD11b+ myeloid cells. These findings are illustrated in figures 1 and 2 shown below.This comparison has been included in the discussion of this manuscript; however the results are not going to be incorporated into this manuscript because they belong to another ongoing study by our research group.

Figure 1: Lung cell infiltrate following SARS-CoV-2 infection. Eight weeks old female B6.Cg-Tg(K18-ACE2)2Prlmn/J mice were  intranasally inoculated with 50 uL of DMEM medium with 2% FBS containing 1x105TCID50 (Tissue Culture Infectious Dose) of SARS-CoV-2 tissue culture) [SARS-CoV-2 - Brazil /SPBR1 / 2020, provided by the Laboratory of Laboratory of Clinical and Molecular Virology, ICB-USP. Seven days post-infection, lungs were collected and processed for flow cytometry analysis of cell infiltrate.

Figure 2: Cytokine production by lung cells following SARS-CoV-2 infection. Eight weeks old female B6.Cg-Tg(K18-ACE2)2Prlmn/J mice were  intranasally inoculated with 50 uL of DMEM medium with 2% FBS containing 1x105TCID50 (Tissue Culture Infectious Dose) of SARS-CoV-2 tissue culture) [SARS-CoV-2 - Brazil /SPBR1 / 2020, provided by the Laboratory of Laboratory of Clinical and Molecular Virology, ICB-USP. Seven days post-infection, lungs were collected and processed and the isolated cells were in vitro stimulated with PMA/Ionomycin for cytokine detection by flow cytometry analysis of cell infiltrate.

The following text was inserted in the discussion section of our manuscript:

The validation of our model as an adequate tool to investigate other procedures to control lung inflammation is supported by another investigation going on in our research group. The histological changes that we found after the instillation of the UV-inactivated SARS-CoV-2 are comparable to the lung inflammation that h-ACE2 mice develop after the active infection. The profile of inflammatory cells eluted from the lung parenchyma is also very similar to the one described in our investigation (Aype et al, unpublished data). This validation is also reinforced by the data described by Bi et al., 2021. These authors developed a model of SARS-CoV-2-induced acute respiratory distress syndrome by intratracheal instillation of formaldeyde-inactivated SARS-CoV-2. Their described histopathological alterations and profile of cells infiltrated in the lungs is also similar to our findings. (Lines 660-669).

2) Although in the Conclusion, the study claims that "...most of the clinical trials done with VitD employed other supplementation routes, mainly the oral one, we suppose that the IN one could be more efficient...". However, oral administration was not tried in the present study, only IP. Thus, such conclusion can be drawn only for IP administration.

Answer

Despite the existence of several and complex differences concerning the IP route in mice (Al Shoyaib et al., 2019)  and the oral one  in humans (Al Shoyaib et al., 2023), both are considered systemic delivery routes. Based on this  and in our higher experience with IP vit D administration   to mice with experimental  encephalomyelitis, this delivery vitD route was used as a way to obtain a systemic distribution of vitD, displaying, therefore, some similarities with the oral route in humans.

The original conclusion was replaced by the paragraph transcribed below:

The results provided  by our investigation suggest a promising potential of vitD delivery  by IN route to control the pulmonary inflammation associated with the presence of SARS-CoV-2 antigens/components in the lungs. Preclinical and clinical further investigations will be essential to determine if these experimental findings can be translated to SARS-CoV-2 infection in humans. (Lines 794-798)

3) It is also claimed in the Discussion that "Theoretically, a higher local concentration of VitD could be more efficient and, possibly, less toxic than systemic administration. Indeed, the higher efficacy of local delivery was confirmed by our findings..." The study did not provide data on local concentration of VitD. The "efficacy of local delivery" is not defined in pharmacokinetics. I cannot see any data on local delivery.

4) Again the statement (in the Discussion) "Considering that the local delivery efficacy was much more impressive than the systemic, we believe that most of VitD effects are locally occurring and are being mediated by some of its well-established immunomodulatory mechanisms." is not supported by any data and the term "local delivery efficiency" is not defined. What is the unit of local delivery efficiency??? And how did you measure that?

Questions three on four Answer below

We agree with the reviewer´s criticism because no tests were done to compare vitD concentrations in  the lungs  of mice treated by IP and IN routes. Additionally, IN administration of vitD is a more recent approach and no tests were done yet to determine its local and systemic concentrations. Considering the reviewer´s criticism and the fact that IN and IP strategies employed distinct protocols, the following paragraph was inserted at the discussion section:

Our initial hypothesis predicting a superior efficacy of IN VitD was based, among other information, on the fact that other lung inflammatory pathologies, such as experimental asthma and rhinitis, were efficiently controlled by local (IN) vitD delivery [59,60]. We also considered the fact that it is increasingly recognized that local synthesis of active vitD is more relevant for many of its immune effects on respiratory diseases than its systemic production (Hansdottir & Monick, 2011). We did not investigate in detail the mechanism by which the IN route, in contrast to the IP one, effectively controlled lung inflammation. We could speculate that the IN protocol, which theoretically allows local availability of vitD during the initial interaction of the virus with pulmonary immune cells, could decrease the intensity of this interaction by, for example, locally decreasing TLR expression. This effect, which has already been demonstrated after exposition of PBMCs to vitD, decreased the production of pro-inflammatory cytokines [Adamczak, 2017]. We could also theorize that local vitD instillation is determining a higher concentration of vitD in the lung draining lymph nodes and in the lungs themselves considering that this is one of the goals of local drug delivery (Labiris & Dolovich, 2003). However, future studies are necessary to measure the local vitD bioavailability and the optimal dose-response kinetics following its IN administration. (Lines 702-717)

5) To draw relevant conclusions and explain the difference in pharmacokinetics of the different administration routes (IN and IP), the measurement of systemic and local concentrations were necessary. The present manuscript does not provide enough data to explain the apparent success of IN administration and I am not convinced that it may be real alternative of oral administration for VitD.

We agree with the reviewer´s opinion that  pharmacokinetics and additional immunological assays will allow a great advance in this area. However, we still believe that the results described here strongly suggest that IN vit D is a procedure  whose use in controlling pulmonary inflammation associated with SARS-CoV-2 deserves to be widely and more deeply investigated. The paragraph below was inserted in the discussion:

Acreditamos que a contribuição mais relevante desta investigação é a prova de conceito de que o IN vitD pode controlar significativamente o processo inflamatório pulmonar desencadeado pela presença local do vírus. Nosso estudo é o primeiro relato de que a via IN pode ser uma alternativa anti-inflamatória adequada para o controle da inflamação pulmonar induzida pelos componentes virais. De fato, estudos futuros são necessários para comparar a eficácia em relação à via oral, para definir uma melhor dose-resposta e também para entender a farmacocinética e a possível redução dos efeitos colaterais sistêmicos associados a ambas as vias de administração. Além disso, já observamos que a inflamação desencadeada pelo SARS-CoV-2 viável se assemelha muito à induzida pelo vírus inativado (manuscrito em preparação). A eficácia da vitD no controle da inflamação durante uma infecção ativa por SARS-CoV-2 precisa de uma investigação futura e cuidadosa e possivelmente exigirá a associação a drogas virucidas. (Linhas 771-782)

Round 2

Reviewer 2 Report

Dear Sirs, 

   please find in attach my comments.  

Author Response

Dear Authors, thank you for your clarification, that made the paper clearer.

Answer:

Dear reviewer number 2, we are extremely grateful for the corrections and recommendations given to our manuscript. We are aware of the time and dedication that this review required from you. Thank you very much.

However, I am still puzzled about your choice to limit research to only females, since CoVID, unlike multiple sclerosis in C57BL/6 mice, affects both sexes equally. I believe that this point must be properly explained or, at the very least, cited in the title.

Answer

We are conscious about the gender differences concerning both, COVID-19 evolution in humans and possibly in experimental infected mice.  Gender differences have also been reported in humans and mice concerning vitamin D metabolism, serum levels and even activity. We certainly agree with you on the relevance of this topic and  the need for clinical and pre-clinical studies to elucidate these aspects in detail. As we are currently unable to carry out these investigations, we would like to change, as you suggested, the title of this manuscript to “Lung inflammation induced by inactivated SARS-CoV-2 in female C57BL/6 mice is controlled by intranasal instillation of vitamin D”. (2-4)

Another concern is the number of some experiments and not the number of animals participating in each group in the same experiment. In this case, I suggest you check very carefully your statements about the significance of the results, as I have pointed out in some cases in the notes.

Answer

Each one of your notes was carefully considered. In addition, the results section was thoroughly read to avoid the valuation of trends.

Personal observation: it makes sense to establish a working model of SARS-CoV-2-like inflammation by the nasal instillation of UV-inactivated SARS-CoV-2, as it is well established that this is the main route of entry of the virus in the human body. Drugs instillation in the nasal cavities may indeed be suitable for local conditions, such as rhinitis (ref. 26) or to reach defined areas of the CNS, in particular those involved in the olfactory pathways; in the case of asthma and, more generally, of pathologies involving the lung parenchyma, which is the target tissue examined in this study and the main target of SARS-CoV-2 virus, the administration by aerosol obtained by ultrasound appears a more suitable strategy to reach the respiratory epithelium, as mentioned in ref 27. So, it might be interesting to test the effects of this type of

administration, which, in a translational perspective, could find application in assisted breathing in human medicine.

Answer

Thanks a lot for sharing your scientific ideas with us.

In fact, we are hopeful that several research groups will get involved with this issue because many practical aspects still need to be solved. In this sense, for example, the need to clarify the moment that vitamin D would be more effective; would be before, during or even after the infection is already established? Would it be useful in the Long COVID?

Notes

Line 20 – bronchoalveolar lavage fluid (BALF) instead of BALF (corrected) Lines 21-22

Line 47 – is instead of as (corrected) Line 48

Line 49 – pathogens instead of pathogen (corrected) Line 50

Line 70-72 – “An accentuated deterioration in pulmonary function was identified a few days later which, coincided with a local infiltration of monocytes, neutrophils and activated T cells.”

This sentence could be reformulated as:

 “An accentuated deterioration in pulmonary function, which coincided with a local infiltration of monocytes, neutrophils and activated T cells, was identified a few days later.” (corrected) Lines 70-72

Line 80 - SARS-CoV-2 spike (S) protein instead of S protein and bacterial lipopolysaccharide (LPS) instead of LPS (corrected) Line 80

Line 83 – BALF instead of bronchoalveolar lavage fluid, as the name in extenso must be cited in the abstract  (corrected) Line 83

Line 112 – (CNS) instead of CNS) (corrected) Line 112

Line 299 and Fig 1B - the percentage variation of lymphocytes in BALF in the groups C, CM, SARS-Cov-2 is … not statistically significant, so the statement: “higher frequency of lymphocytes” has no basis.

Answer

Yes,  the increase in the percentage of lymphocytes in this figure  was discreet; we therefore  replaced the original phrase by: The percentage alterations observed in the SARS-CoV-2 group included only a significant decrease in macrophges and a significant increase in neutrophils.  Lines 296-298

Line 303 and Fig 1B – The percentual increase in the number of the macrophages in Control and Culture medium groups with respect to SARS-Cov-2 group is statistically significant and, in my opinion, should be discussed and explained. Moreover, it contradicts the statement from line 300 to line 301.

Answer

We do not see this diference in macrophage´s percentage (Figure 1B) as an indication that there is an increase in macrophages in these two control groups. On the contrary, we see this result as an indication that the percentage of macrophages in the SARS-CoV-2 group is lower than in the two control groups because there is a migration, and therefore an increment of other cell types as lymphocytes (discreet) and PMNs (significant) in the lungs. Even though the proportion of macrophages is smaller in SARS-CoV-2, the total number of this cell type is almost double in comparison to the two control groups (Figure 1A). The following paragraph was added to the results section:

This lower percentage of macrophages in the SARS-CoV-2 group, in comparison to the control groups, indicates an increment of other cell types as lymphocytes (discreet) and PMNs (significant) associated to the cellular influx to the lungs triggered by the virus. Even though the proportion of macrophages is smaller, the total number of this cell type is almost double in the SARS-CoV-2 group, in comparison to the  control groups (Figure 1A). (Lines 298-303)

Line 361- 2 and Fig. 2H, 2J: the increase of the total number of monocytes and monocytes-derived macrophages in SARS-CoV-2 group in comparison to the control group is not significant. (corrected) Lines 363-368

Line 366-8: Since this statement is not supported by data, I suggest adding the phrase "data not shown".  (Corrected) Lines 368-370

Line 400-1: Please, cite the bibliography to support this statement. (Corrected) Line 404

Line 402: please, add “inactivated” before SARS-CoV-2 (corrected) Line 406)

Line 405: effect instead of effet (corrected) Line 403

Line 437; Please add: In lungs of mice IN challenged with UV inactivated SARS-CoV-2

Line 473-480 and Fig. 5B-5C: why the authors didn’t check the total number and the frequency of CD45+ cells infiltrating the lung parenchyma of mice intraperitoneally treated with inactivated SARS-CoV-2? This data could be important in order to explain the results showed in the hystological sections.

 Since the IP treatment did not alter the inflammatory infiltrate in the lung sections, in this phase of the investigation and due to practical constraints regarding animal disponibility, we opted for the inclusion of more animals in the VitD-treated group than to keep the IP group. We certainly agree that we lost the opportunity, at this moment, to understand better this differential efficacy of VitD depending upon the route.

Line 481, 512, 515, 520….549: please, add IN before VitD. (corrected) Lines 483, 512, 517, 520 e 552

Line 535 and Fig 6I: The total numbers of IFN-γ producing ILCs variation is not statistically significant. (corrected) Line 537

Line 774: in my opinion, the adjective "alternative" is excessive; perhaps better "additional" or "adjuvant".

Answer

As replacing the original “alternative” by “adjuvant” or “additional” did not seem very clear, we opted to reduce the impact of the information substituting the original phrase by a new one. Please, see the new phrase below:

Our study seems to be the first report suggesting that IN vitD administration has the potential to control inflammation induced by viral components.  (776-778)

Reviewer 3 Report

The answers are acceptable.

Author Response

Dear reviewer number 3, we are extremely grateful for the corrections and recommendations given to our manuscript. We are aware of the time and dedication that this review required from you. Thank you very much.